# Investigating the potential effectiveness of earthquake early warning across Europe

Gemma Cremen [1✉], Carmine Galasso[1,2] & Elisa Zuccolo[3]

Here we assess the potential implementation of earthquake early warning (EEW) across Europe, where there is a clear need for measures that mitigate seismic risk. EEW systems consist of seismic networks and mathematical models/algorithms capable of real-time data telemetry that alert stakeholders (e.g., civil-protection authorities, the public) to an earthquake's nucleation seconds before shaking occurs at target sites. During this time, actions can be taken that might decrease detrimental impacts. We investigate distributions of EEW lead times available across various parts of the Euro-Mediterranean region, based on seismicity models and seismic network density. We then determine the potential usefulness of these times for EEW purposes by defining their spatial relationship with population exposure, seismic hazard, and an alert accuracy proxy, using well-established earthquake-engineering tools for measuring the impacts of earthquakes. Our mapped feasibility results show that, under certain conditions, EEW could be effective for some parts of Europe.

[1] University College London, London WC1E 6BT, UK. [2] Scuola Universitaria Superiore (IUSS) Pavia, 27100 Pavia, Italy. [3] European Centre for Training and Research in Earthquake Engineering (EUCENTRE), Department of Risk Scenarios, 27100 Pavia, Italy. ✉email: g.cremen@ucl.ac.uk

Earthquake early warning (EEW) systems are a relatively recent innovation in earthquake-induced disaster risk reduction and resilience promotion[1]. They consist of seismic sensor networks and mathematical models/algorithms that are designed to process and disseminate real-time information about ongoing earthquakes. The resulting alert messages enable various stakeholders (e.g., individuals, communities, governments, businesses) located at distance to take timely measures for reducing the likelihood of damage or loss before shaking reaches them[2]. Examples of important risk-mitigation actions that can be taken in the short warning time provided by EEW systems (typically seconds) include: (1) Performing "drop, cover and hold on" (DCHO)[3] or moving to a safer location (either within a building or outside), to avoid injuries; (2) slowing down high-speed trains, to reduce accidents[4]; (3) shutting off gas pipelines, to prevent fires[5]; and (4) switching signals to stop vehicles from entering vulnerable infrastructure components (such as bridges), to avoid fatalities[6]. This list accounts for merely a small number of the vast array of critical applications that can benefit from EEW[7], and interested readers are referred to Wald[8] for a more thorough discussion on this issue.

The process of EEW typically involves up to five main steps: (1) detecting an earthquake; (2) estimating its location; (3) estimating its magnitude; (4) estimating the ground motion at target sites; and (5) using all of the information collected to decide whether (or not) to trigger an alarm. EEW systems may be broadly categorised as "regional", "on-site", or "hybrid", depending on their approach to the first four steps mentioned above. This study exclusively focuses on regional systems, which consist of seismic station networks installed within the expected epicentral/high seismicity area that record the necessary information for estimating the parameters of Steps 1, 2 and 3. The source parameter estimates of Steps 2 and 3 are then used to predict ground shaking (Step 4) at target sites located further away from the fault rupture[9].

A number of studies have previously explored the feasibility/potential of EEW in different parts of the world, including France[10], Italy[11–13], Spain[14], Portugal[15], Turkey[16], Japan[17], California[18,19], Hawaii[20], the New Madrid Seismic Zone[21], and Kyrgyzstan[22]. Regional EEW systems are presently operating in nine countries (including USA, Mexico, and Japan), and have been tested for application in a further 13[23]. The only European countries with current government-supported operational EEW systems are Romania[24] and Turkey[25] (the Android Earthquake Alert System, which uses Android phones to issue and receive early warnings[26–29], has also recently been launched in Greece as well as Turkey[30]), despite a strong need to develop effective measures for mitigating seismic risk across many parts of the

continent[31]; EEW could potentially contribute towards reducing the more than 20 billion of European gross domestic product (GDP) that is affected annually by earthquakes (on average)[32].

In this study, we investigate the feasibility of EEW application in the Euro-Mediterranean area. In particular, we focus on EEW lead time (i.e., the time between the delivery of an EEW alert and the arrival of shaking at target sites). We compute probabilistic distributions of lead times available for various seismicity scenarios in high-hazard areas across the continent, using a finite-difference travel-time algorithm. We also explicitly quantify the potential effectiveness of these times in the context of EEW, by establishing their spatial relationship with values of proxy measures for earthquake impact and alert accuracy. This work significantly advances the state-of-the-art established by aforementioned studies for a number of reasons. It examines EEW feasibility on a much larger (i.e., continental level) scale by combining EEW methods, models, and tools in a harmonised framework across Europe. Furthermore, we introduce a feasibility metric that enables identification of priority regions for further, more refined EEW feasibility analyses and/or actual investment in EEW systems for targeted end users. This study therefore offers a unique trans-national perspective on the potential of EEW that is relevant for intergovernmental bodies—such as the International Search and Rescue Advisory Group of the United Nations[33]—who may be interested in leveraging the technology. It also provides valuable new insights on the possible benefits/limitations of EEW for regions (e.g., Iceland and Georgia) that have not recently experienced large earthquakes, but are likely to do so in the future.

## Results

**European seismic station density**. We conduct a preliminary feasibility study for EEW across the European region, by considering the availability of its most fundamental component, i.e., seismic station networks on which the early seismic signals could be detected/recorded for rapid event characterisation. Figure 1a displays a map of permanent European broadband and strong-motion seismic stations (2377 stations in total). It can be seen from Fig. 1b that ~45% of interstation distances are less than 20 km and almost all interstation distances are within 100 km.

**Lead-time mapping for high-hazard areas**. We now focus on crustal point sources associated with large seismic hazard of engineering significance, which we define as those for which the event with a recurrence interval of 500 years is at least $M_w$ 6.5 (see Fig. 2a and "Methods" section). For each of these area sources, we calculate potential lead times (i.e., times between

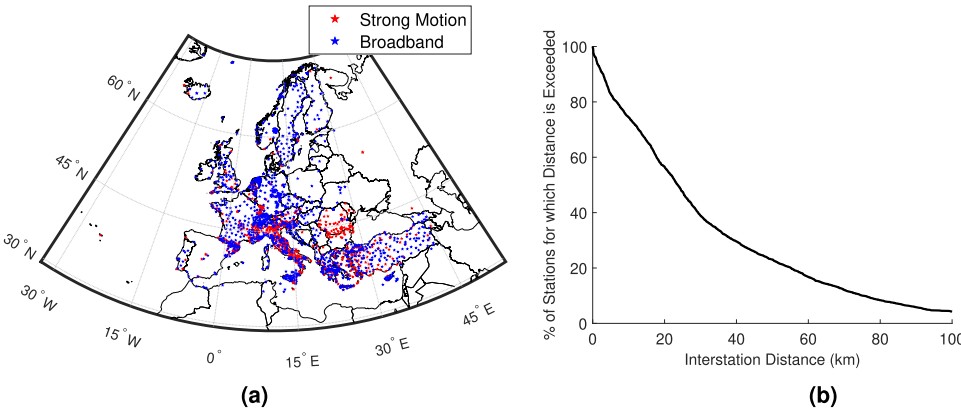

**Fig. 1 Examining seismic station coverage across Europe. a** Map of European seismic stations considered and **b** distribution of interstation distances.

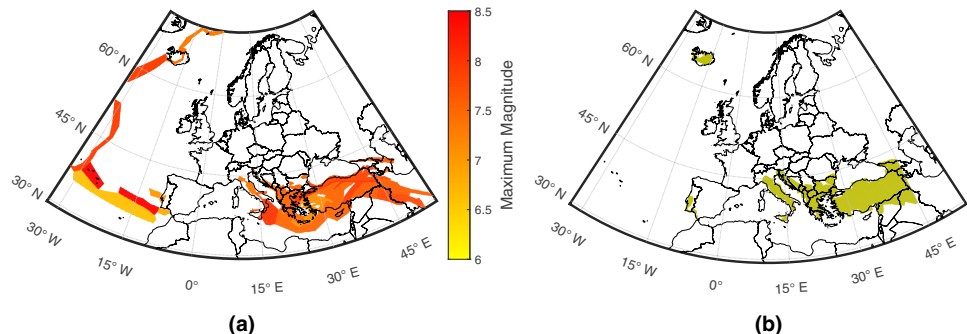

**Fig. 2 Input data for lead-time mapping calculations. a** Seismic sources (colour coded in accordance with corresponding modal maximum magnitude values from the seismic hazard model) and **b** target sites examined for lead-time calculations.

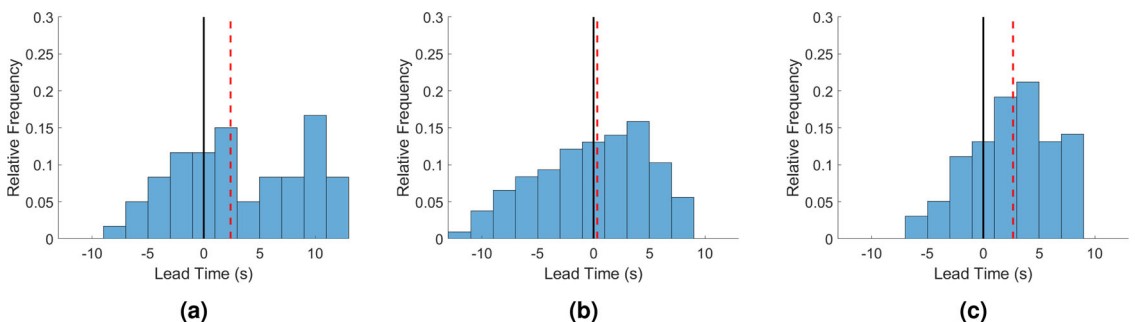

**Fig. 3 Distributions of lead times for target sites in three European cities. a** Site in Naples, **b** site in Izmir, and **c** site in Athens. Note that the red dashed lines indicate the corresponding median lead times and the black solid lines denote the positive lead-time threshold.

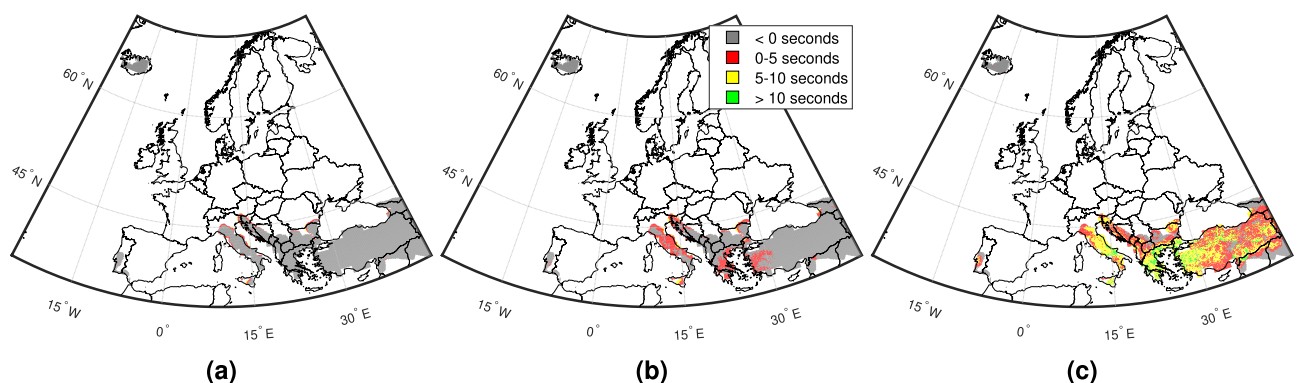

**Fig. 4 Lead-time mapping across all examined target sites. a** Minimum lead times, **b** median lead times, and **c** maximum lead times.

EEW alert issuance and the occurrence of shaking) at target sites where the predicted median peak ground acceleration (PGA) associated with 500-year recurrence-interval events exceeds 0.05 g (see Fig. 2b and "Methods" section), which is a commonly used threshold value for moderate earthquake shaking in several engineering applications, including seismic design aimed at life-safety performance[34–36]. These calculations incorporate a magnitude-dependent delay interval that captures the time taken to compute characteristics of the ongoing event and to complete a state-of-the-art real-time data telemetry process (see "Methods" section).

Figure 3 displays histograms of lead times computed at selected target sites in three cities covered by the study area, i.e., Naples, Izmir, and Athens, due to the area sources that comply with the previously outlined criteria. It can be seen that the majority of lead times are positive for the selected sites in Naples

and Athens, whereas there is a reasonably even distribution of both positive and negative lead times at the Izmir site. The median lead times for the sites are 2.4 s (Naples), 0.3 s (Izmir), and 2.7 s (Athens), while the standard deviations of the times (in the same order) are 5.8, 4.9, and 3.8; these uncertainties are significant, and are partly explained by the large variation in source-to-site distances for a given site.

Figure 4 contains maps displaying the following three summary statistics for all affected target sites across the continent: (1) lowest computed lead time (i.e., "worst case scenario"), henceforth referred to as "minimum lead time"; (2) median computed lead time; and (3) largest computed lead time (i.e., "best case scenario"), herein referred to as "maximum lead time". Note that negative lead times correspond to blind zones, where no warning is received before shaking occurs. Of all target sites examined, 3% have positive minimum lead times

(<1% between 5 and 10 s, and the remainder less than 5 s); 18% have positive median lead times (1% between 5 and 10 s, and approximately 16% less than 5 s); and 79% have positive maximum lead times (8% greater than 10 s, 37% between 5 and 10 s, and 34% less than 5 s). The maximum lead time achieved across all target sites examined is 17.2 s (near Sorgun, central Turkey). Target sites with the longest overall median lead times are mainly found in Italy, Greece and Turkey, which are characterised by some of the strongest seismicity in Europe. Target sites with the shortest lead times are located in Iraq, Georgia, and Russia. Table 1 provides a summary of potential risk-reducing actions that can be carried out for the various ranges of lead time investigated.

*Lead-time sensitivity analyses.* We now examine variations in lead times that result from modifying certain assumed inputs of the previous calculations. We first determine lead times for $M_w$ 5 events at the previously considered point sources, focusing on target sites where these earthquakes produce median predicted PGA greater than 0.05 g (Fig. 5). Such moderate earthquakes can sometimes have notable consequences[8], so it is important to understand whether EEW systems could successfully operate for these events. No target sites have positive maximum lead times in this case.

We next determine lead times for the same earthquakes considered in the original calculation that produce a PGA of at least 0.1 g at a given target site, to account for stakeholders who may only wish to trigger EEW alerts in the case of strong shaking[37] (see Fig. 6). Less than 1% of these sites have positive minimum or median lead times (which are all smaller than 5 s, in both cases), and 19% have positive maximum lead times (<1% between 5 and 10 s, and the remainder less than 5 s).

**Quantifying the effectiveness of computed lead times.** We examine the potential usefulness of the original calculated lead times for EEW purposes, by defining their spatial relationship with ambient (average day/night) population distributions and the average seismic intensity across all events with a recurrence interval of 500 years that produce a PGA greater than or equal to 0.05 g at the affected site (see "Methods" section for details). Population often acts as a proxy for the exposure (i.e., the value at risk) of the built environment/assets in earthquake engineering and risk modelling applications[38]. Seismic intensity describes the effect of earthquake ground shaking on the built environment and communities[39,40]. We use the European Macroseismic Scale (EMS)-98 seismic intensity scale[40], which is specifically designed for European countries.

Ninety-eight percent of the total ambient population surrounding the examined target sites are affected by average EMS-98 values between V ("Strong"; e.g., top-heavy objects topple over) and VII ("Damaging"; e.g., many objects fall from shelves and there is some wall damage). Figure 7 indicates that ~30% of the ambient population are affected by EMS-98 values between V and VI ("Slightly damaging"; e.g., objects fall from walls and there is some damage to plaster), while ~68% are affected by values between VI and VII. Five percent of the ambient population affected by average intensities between V and VI have maximum lead times greater than 10 s, while 22% have negative maximum lead times (i.e., they are located in the "blind zone"). Thirty-two percent of this population have positive median lead times, and

**Table 1 Possible risk-mitigation actions that can be taken by various stakeholders for different lengths of EEW lead time (adapted from previous works[3,15,71-73]).**

| Lead time range (s) | Possible actions |
| --- | --- |
| 0–5 | • Stopping traffic (i.e., turning lights red)<br>• Switching on semi-active control systems for structures |
| 5–10 | • Performing DCHO<br>• Stopping elevators at the nearest floor and opening doors<br>• Shutting off gas supplies<br>• Shutting down computers and related equipment<br>• Evacuating the ground floor of buildings |
| >10 | • Shutting down industrial equipment<br>• Controlling production lines<br>• Directing traffic away from underpasses<br>• Stopping surgical procedures<br>• Removing vehicles from garages |

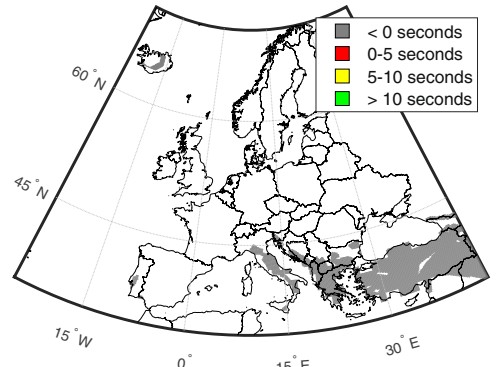

**Fig. 5** Lead-time mapping for $M_w$ 5 events.

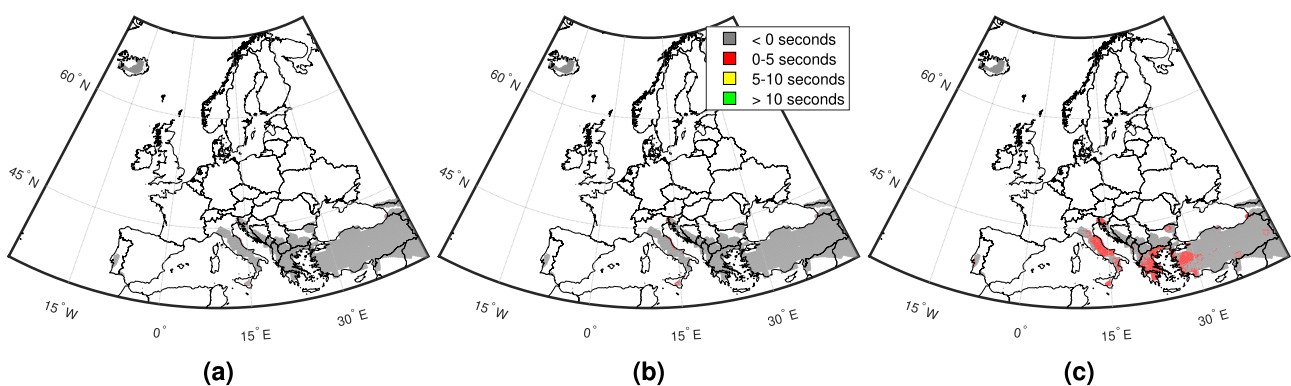

**Fig. 6 Lead-time mapping for a 0.1 g EEW alert threshold. a** Minimum lead times, **b** median lead times, and **c** maximum lead times.

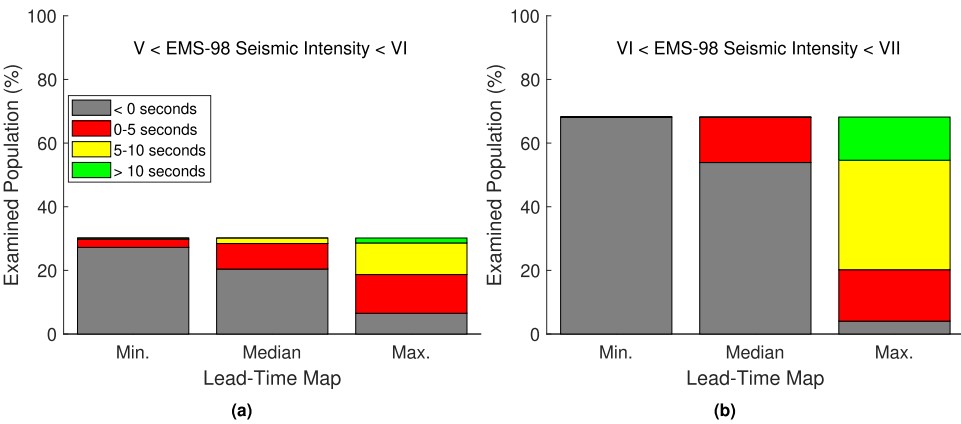

**Fig. 7 Examining the potential effectiveness of calculated lead times for EEW.** Average EMS-98 macroseismic intensities experienced by the affected ambient population during events with a recurrence interval of 500 years that resulted in at least 0.05 g PGA at the associated target site, categorised by the corresponding times of the maps presented in Fig. 4. Note that seismic intensities V, VI, and VII denote "strong", "slightly damaging", and "damaging" events, respectively.

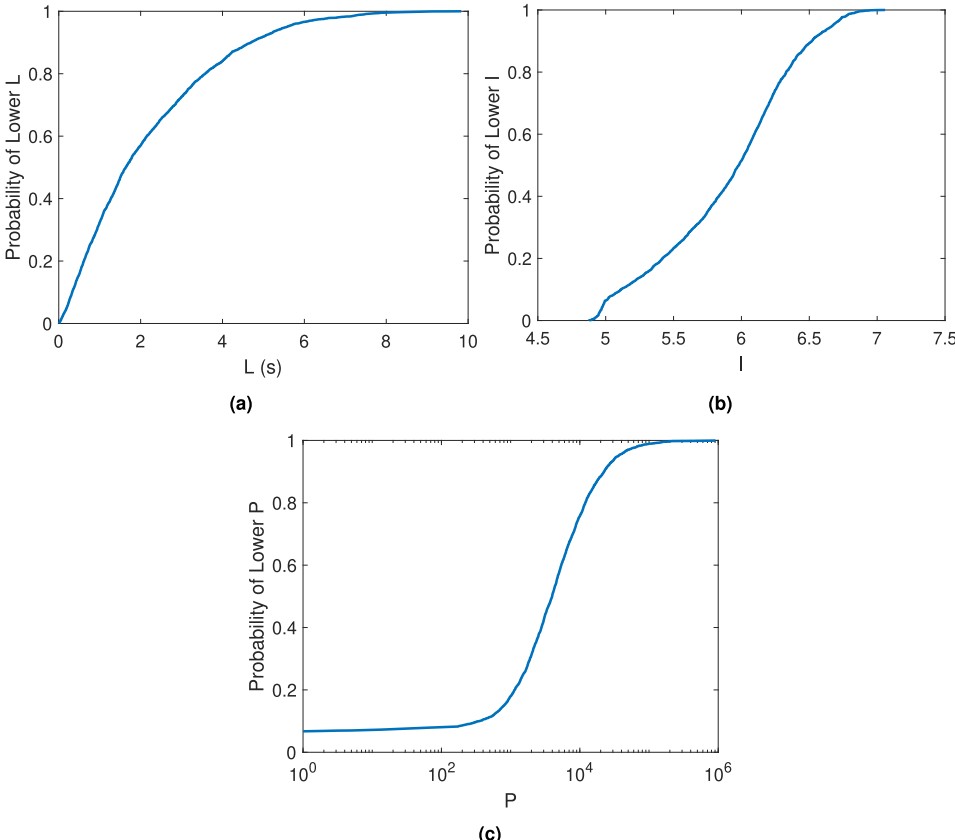

**Fig. 8 Deriving the EEW relative feasibility index.** Empirical cumulative distribution functions of **a** median lead time ($L$), **b** average seismic intensity ($I$), and **c** ambient population ($P$).

10% have positive minimum lead times. Twenty percent of the ambient population affected by average intensities between VI and VII have maximum lead times greater than 10 s, while 6% are located in the "blind zone". Twenty-one percent of this population have positive median lead times, and less than 1% have positive minimum lead times.

*EEW feasibility calculation.* We combine estimates of median lead time ($L$), average seismic intensity ($I$), and affected ambient population ($P$) into a single metric of EEW feasibility, termed the EEW relative feasibility index, which ranges from 0 to 1 (see Eq. (3) of

"Methods" section for details). Higher values of this index correspond to key characteristics that maximise the effectiveness of an EEW system[41], i.e.,: (1) longer lead times; (2) higher potential for shaking causing losses that can be avoided with EEW; and (3) larger affected populations. They therefore indicate greater EEW feasibility for a given target site. For context, Fig. 8 provides the empirical cumulative distribution functions (ECDF) of $L$, $P$, and $I$ that are used to derive the index. Note that the purpose of the index is to identify the most feasible regions for EEW, regardless of the extent to which their feasibility differs to that of less feasible regions (and therefore the steepness of the underlying ECDF). This approach is

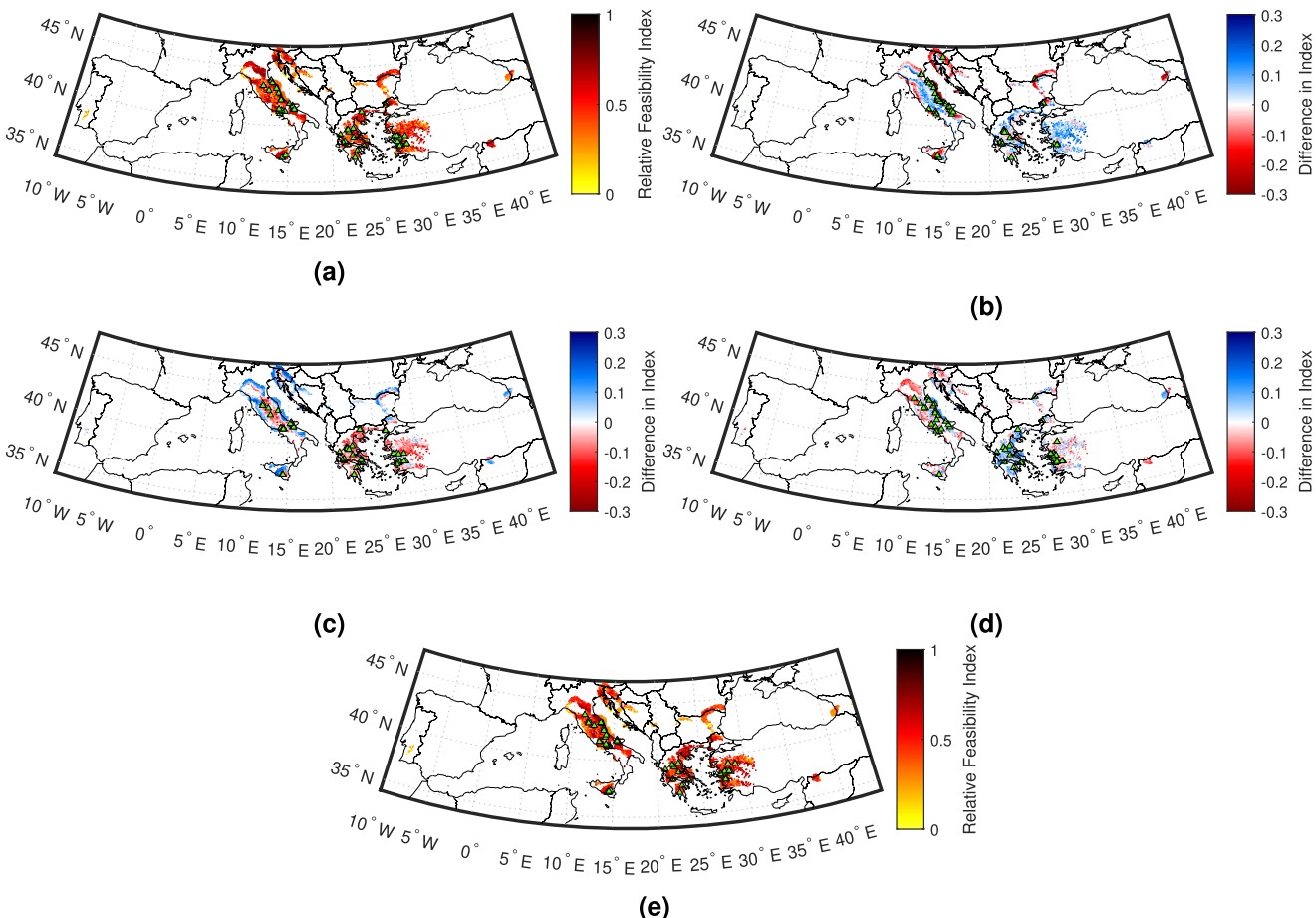

**Fig. 9 Relative feasibility index mapping across examined target sites.** Indices for **a** the case in which lead time, intensity, and population are equally weighted by a stakeholder, as well as differences for cases in which **b** lead time, **c** seismic intensity, and **d** population are respectively weighted three times more than both other variables. Also shown are **e** equally weighted indices modified in line with the relative number of correctly issued alerts (for a 0.05 g alert threshold)[19]. Note that for **b**–**d**, red colours indicate an increase in the index relative to **a** and blue colours indicate a relative decrease. Green triangles indicate target sites with one of the 50 largest indices for each case.

consistent with many multi-criteria decision-making tools, including the Technique for Order of Preference by Similarity to Ideal Solution[42] and multi-attribute utility theory[43].

It can be seen from Eq. (3) that the index accommodates a user-defined weight for each measurement, to account for stakeholder preferences and priorities towards each feature of feasibility. Figure 9 includes EEW Relative Feasibility Index mapping of all target sites with positive median lead time, for the equally weighted case (i.e., $w_P = w_I = w_L = 0.333$) and for cases where one variable (e.g., lead time) is weighted three times more than the other two (e.g., $w_L = 0.6$, and $w_P = w_I = 0.2$). Note that a site associated with the 10th percentile value of positive $L$, the 20th percentile value of $I$, and the 40th percentile value of $P$ would yield the following relative feasibility indices for the different examined weighting strategies: 0.23 ($w_P = w_I = w_L = 0.333$), 0.18 ($w_L = 0.6$, and $w_P = w_I = 0.2$), 0.22 ($w_I = 0.6$, and $w_L = w_P = 0.2$), and 0.3 ($w_P = 0.6$, and $w_L = w_I = 0.2$). In contrast, for the same respective weighting strategies, a site associated with the 90th percentile value of positive $L$, the 80th percentile value of $I$, and the 60th percentile value of $P$ would produce relative feasibility indices of 0.77, 0.82, 0.78, and 0.7. Also highlighted in each subplot of Fig. 9 are the fifty target sites with the largest index values for the corresponding case. For all cases, the countries containing all (or almost all) of the sites with the fifty largest feasibility indices are Italy, Turkey, and Greece. However, both the locations and the number of sites per country differ between cases. Relative to the equally weighted case, target

sites with the largest increase and decrease in feasibility index for the case where lead time is the most weighted variable are located in Georgia and Turkey respectively, target sites with the largest increase and decrease in this value for the case where seismic intensity is the most weighted variable are located in Greece and Georgia respectively, and target sites with the largest increase and decrease in feasibility index for the case where population is the most weighted variable are respectively located in Italy and Greece.

Finally, we investigate the impact of alert accuracy (i.e., the ability of the system not to miss alarms or provide false warnings) on EEW feasibility. We specifically adopt the approach of Minson et al.[19], which examines the forecasting capability of EEW in terms of ground motion prediction accuracy for a set of known source parameters. We randomly sample PGA values at each site for a series of earthquakes across nearby sources, assuming that an alert is issued if the corresponding median predicted PGA exceeds 0.05 g. The relative feasibility indices of Fig. 9a are then modified in line with the relative proportion of correctly issued alerts to produce Fig. 9e; see Eqs. (4) and (5) of "Methods" section for details. Alert accuracy causes the largest feasibility increase and decrease at sites in Slovenia and Romania, respectively. However, Turkey, Italy, and Greece still maintain the largest feasibility. Although alert accuracy is highly dependent on the selected threshold[19], it is important to note that the top three countries for EEW feasibility do not change if the triggering PGA

is instead set to 0.02 g, 0.10 g, or 0.20 g, i.e., the three exceedance values examined in Minson et al.[19] for California.

## Discussion

This study has examined the relative feasibility of EEW for Europe. We initially analysed the density of seismic station coverage across the continent. We found that almost half of inter-station distances are less than 20 km, which corresponds with the distance limit recommended for optimum EEW performance in previous work[41]. These findings are a preliminary signal that there is some potential for operational EEW across the continent. Our detailed relative feasibility analysis focused on Euro-Mediterranean regions affected by significant seismic hazard. This indicated that the viability of EEW in Europe is highly dependent on the magnitude of the ongoing event and the threshold PGA at which an EEW alert is issued at a target site. For example, it was determined that 45% of the examined target sites could benefit from lead times in a "best case scenario" that are long enough to accommodate some important risk inter-vention actions—such as the shutting off of gas supplies and the evacuation of ground floors—if the magnitude is large (i.e., at least $M_w$ 6.5) and the threshold for EEW alert triggering is 0.05 g, but this proportion reduces to less than 1% if the triggering PGA is instead set to 0.1g, and no sites benefit from these long lead times for a $M_w$ 5 event. Eighteen percent of all examined target sites have a 50% chance of receiving an EEW alert that allows time for at least some automatic actions (such as the switching of a traffic light), for a large magnitude event and a 0.05 g triggering threshold, but no (or almost no) sites have sufficient median lead time to facilitate these types of measures if the magnitude is reduced to $M_w$ 5, and/or the triggering threshold is instead set to 0.1g. In a "worst case scenario", large magnitude earthquakes and a 0.05 g triggering threshold are the only examined conditions that produce some positive EEW lead time at more than 0.25% of sites. In summary, the ultimate success of European EEW (from a functionality standpoint) will be dictated by the practical conditions of its usage and the underlying seismotectonic setting(s). We found that the longest overall lead times mainly occur at sites in Italy, Greece, and Turkey. Areas associated with the shortest overall median lead times, and therefore where the fea-sibility of EEW could be improved through increased seismic station density, are northern Iraq, north-western Georgia, and southern Russia.

We further contextualised the significance of the lead times by combining them with spatial distributions of two proxies often used to measure the effects of an earthquake in earthquake engineering (i.e., population and seismic intensity). We found that almost all (i.e., ~98%) of the affected ambient population are exposed to average seismic intensities from large earthquakes at nearby seismic sources that at least result in some falling objects. This suggests, for example, that EEW could help to protect against injuries through DCHO, evacuation or other means. Fifteen percent of these people have greater than 10 s of lead time in a "best case scenario", which enables them to carry out major preventive actions, such as the shutting down of industrial equipment. A notable amount of the population (25%) have positive lead times from events at 50% of relevant sources, while ~4% have some lead time in a "worst case scenario". These findings indicate that European EEW could be useful for miti-gating the effects of large events on exposed populations.

Finally, we translated the aforementioned features (i.e., (1) lead time; (2) average seismic intensity from large earthquakes at nearby sources; and (3) ambient population affected) into an indicator for measuring relative EEW feasibility at a given target site that also accounts for stakeholder-specific preferences (weights). While there was some variation in the results obtained for different weighting strategies, all maps indicated that Turkey, Italy, and Greece contain all (or almost all) of the target sites with the highest relative EEW feasibility. We additionally examined the impact of alert accuracy on the equally weighted relative feasibility map, and found that the same three countries still demonstrated the largest relative feasibility, regardless of the alert threshold considered[19].

In particular, the computed relative feasibility indices suggest that an expansion of permanent EEW efforts in Turkey beyond Istanbul (by upgrading the hardware and software of existing strong-motion/broadband stations and networks for real-time data processing and telemetry capabilities) could be appro-priate, supporting the recently launched Android Earthquake Alert System in the country. The promising results of the relative feasibility mapping for Italy and Greece are particularly notable, since neither has a current permanently operational EEW system (although Greece is now also benefiting from the Android Earthquake Alert System). We ultimately conclude that this work provides evidence to suggest that some parts of Europe could benefit from EEW as a helpful supplemental tool for supporting earthquake-related disaster risk reduction[8] but the extent of its effectiveness would be highly sensitive to the size of targeted events and the threshold at which an alert is triggered.

It is important to note that there are some limitations/sim-plifying assumptions associated with this work that warrant comment. Firstly, we leveraged an international database to obtain details on seismic stations across the continent (see "Methods" section). This approach may not have completely captured all stations across Europe and our calculations may have underestimated actual lead times in some cases; for the actual design/implementation of EEW systems in any region, an exhaustive search of local databases would be critical to produce detailed and accurate lead-time estimates. However, exploiting local seismic station databases in this study could have created an unfair bias against countries/regions that do not provide/store this type of information and may have introduced discrepancies in the quality of information used. In fact, all of the data employed in this work (including those related to seismic hazard and population) are from consistent and open high-level sources to reflect the broad geographical extent of the study and ensure the results are fully replicable. Secondly, it is assumed that the considered seismic stations are (or could be) capable of being used for early warning purposes (i.e., they have or could have adequate data acquisition/transmission systems, real-time com-munication capability, robust dissemination methods, power supply systems, etc.[10,20]), which may be an over-simplification[1]. The times considered in Table 1 for taking prescribed actions during an EEW alert may be longer in practical cases, given that human reaction latencies have not yet been well-established in this context[8]. We used a 1-D velocity model in the travel-time algorithm, which does not capture lateral variations in the earth's structure. Our detailed EEW feasibility analysis only accounted for crustal seismic sources, thereby yielding conservative lead times for target sites that would additionally be affected by the deeper seismicity of subduction zones in the Central and Eastern Mediterranean Sea. It therefore also neglected the seismicity of the Vrancea region in Romania[44], which has significant asso-ciated hazard[45]; examination of this region is not crucial in the context of our study however, given that it already has an operational EEW system[24,46]. In any case, preliminary investi-gations indicate that the conclusions of the relative feasibility mapping do not strongly depend on the accuracy of the lead-time calculations; using interpolated values of interstation dis-tance (see "Methods" section for details on this metric) as a

proxy for lead times in the feasibility index (where smaller distances indicate higher relative feasibility, in line with the findings of Kuyuk and Allen[41]) still produces the largest index values in Turkey, Italy and Greece. To maintain a uniform approach for the entire examined region, the considered seismicity scenarios were defined using an area source model (see "Methods" section), which assumes a uniform occurrence of earthquakes as point sources; thus, the resulting calculations of hazard near faults (large seismogenic sources) may not be completely realistic[47]. Our approach to quantifying alert accuracy only considered the variability of a ground motion model (GMM)[19]. Precisely characterising warning accuracy would involve more detailed analysis with the specific algorithms of operational EEW platforms, including the quantification and propagation of uncertainties at each step of the calculations. This type of examination was carried out for select testbed sites across Europe in previous studies by the same authors[48,49]. It is outside the scope of this paper, given the continent-wide extent of the study (i.e., it is likely that different EEW algorithms would suit different regions) and the fact that this work is foremost an investigation of feasibility. Finally, we did not consider the economic value of EEW, i.e., the costs required to build and maintain EEW systems compared to the monetary savings they provide through avoided damage[50]. Despite these constraints, this study nevertheless represents a first attempt to comprehensively quantify potential EEW effectiveness on a continental scale and to identify priority regions for more detailed EEW feasibility analyses/investment in EEW implementation.

## Methods

### Data descriptions

*Seismic stations*. We use current seismic station locations in this work (and thus account for the geometrical characteristics of the network, assuming that necessary hardware/software upgrades for EEW are possible), in line with previous studies that have examined EEW feasibility[12,51]. Station coordinates are obtained using the Incorporated Research Institutions for Seismology (IRIS) Google map (GMAP) station mapping service (http://ds.iris.edu/gmap/). We consider all permanent strong-motion and broadband stations between −26° and 45° longitude, and 34° and 72° latitude.

*Seismic sources*. We use the area source model of the 2013 European Seismic Hazard Model, which accounts for crustal seismicity with depth ≤40 km[52,53]. To define seismic sources, we discretise the model into 0.1° × 0.1° cells. We specifically make use of the depth, maximum magnitude, style-of-faulting, and Gutenberg–Richter $a,b$ parameters from the model. Each source is assumed to be characterised by all parameter values associated with the corresponding area source zone. We use the values associated with the highest weight in the logic tree, where applicable, and average depth values for stable continental regions. The moment magnitude of the event with a recurrence interval of 500 years for a given source ($m$) satisfies the following equation:

$$\lambda_m - \lambda_{m_{max}} = 0.002 \tag{1}$$

where $\lambda_m$ is the annual rate of earthquakes with magnitude greater than $m$, according to the Gutenberg–Richter magnitude-frequency relationship[54], and $m_{max}$ is the modal maximum magnitude for the given source. We focus on the 37,869 sources for which $m \geq 6.5$. The catalogue generated to quantify alert accuracy consists of 1,000 earthquakes per source that are Gutenberg–Richter distributed and have uniform annual rates of occurrence (from Eq. (1)) between 0 and 1. Predictions of PGA and peak ground velocity (PGV) associated with all events are computed/sampled using the Joyner–Boore distance version of the Akkar et al. GMM[55]. (We compute Joyner–Boore distances from epicentral distances, using the adjustment factors of Thompson and Worden[56] for the style-of-faulting and tectonic setting of the associated seismic source). The site amplification input to the GMM is the shear wave velocity in the uppermost 30 m, which is estimated at each target site from a topographic slope map[57].

*Target sites*. Target sites are equivalent to all land-based seismic sources (i.e. those without a water layer at or above zero-elevation in the corresponding 1-D velocity profile; see "Travel times" section), located within the same coordinate boundaries as the seismic stations.

### Seismic station density

Interstation distance for a given seismic station is the average distance to the closest three stations.

### Lead-time modelling

*Travel times*. We use the travel-time algorithm of the open-source NonLinLoc software package (http://alomax.free.fr/nlloc/)[58]. This method calculates first arrival travel times for the nodes of a spatial grid using the Eikonal finite-difference scheme of Podvin and Lecomte[59], which is an approximation of Huygen's principle[60]. We use a grid spacing of 10 km in all directions, and incorporate a normally distributed zero-mean timing error with 0.2 variance. Both source-to-site and source-to-station travel times are calculated using 1-D velocity profiles from the CRUST 1.0 velocity model[61], at the location of the target site. Note that travel times are computed to zero-elevation at the target site.

*Lead-time calculation*. The lead time (in seconds) for target site $j$ due to an event at a given seismic source $a$ is calculated as follows:

$$LT_j = TT^s_{a,j} - TT^p_{a,st_3} - \delta_m - \delta_t \tag{2}$$

where $TT^s_{a,j}$ is the S-wave arrival time at $j$, and $TT^p_{a,st_3}$ is the P-wave arrival time at the third closest station to the source. We account for the triggering of three stations, as it is the minimum required for many popular regional EEW algorithms to report reliable source parameter estimates[62–64]. $\delta_m$ represents the time required to compute the magnitude of the ongoing event, and is assumed to equal 3 s for $M_w < 6.5$, 4 s for $6.5 \leq M_w < 7$, 12 s for $7 \leq M_w < 7.5$, and 20 s for $M_w \geq 7.5$[65]. The value of $\delta_m$ used for a given area source is based on the magnitude of the event with a recurrence interval of 500 years (except when $M_w$ 5 events are examined, in which case $\delta_m$ is uniformly set to 3 s). Note that the relatively large $\delta_m$ values for magnitudes greater than or equal to 7 require an implicit assumption that the underlying EEW algorithm is capable of filtering out polluting S-waves from long P-wave windows. The validity of this assumption does not significantly affect the outcomes of the study, however; removing from the analyses sources for which $\delta_m \geq 12$ and the nearest station is less than 80 km does not change the conclusions of the work. $\delta_t$ captures data telemetry delays, which are idealistically assumed to comprise 1 s for data transmission and 1 s for issuing the warning message[66–68].

### Lead-time effectiveness modelling

Seismic intensities are calculated from the bilinear equations for EMS-98 macroseismic intensity developed by Masi et al.[69], using median PGV predictions (see "Seismic sources" section). Population data are obtained from the 2018 LandScan database[70], which contains global ambient population distributions at a 30″ × 30″ spatial resolution. Each target site is assigned the aggregated population across all LandScan grid points closest to it.

### EEW feasibility modelling

The relative feasibility index measure for target site $j$ ($RF_j$) considers its associated values of median lead-time ($L$), average seismic intensity ($I$), and ambient population ($P$):

$$RF_j = F_L(l_j) \times w_L + F_I(i_j) \times w_I + F_P(p_j) \times w_P \tag{3}$$

where $F_X(x_k)$ is the ECDF of $X$ (across all examined target sites with positive median lead time) evaluated at target site $k$, and $w_X$ is the stakeholder-assigned weight for $X$ (note that $w_P + w_I + w_L = 1$). Each $F_X(.)$ function ranks the sites based on the underlying metric (i.e., $L$, $I$ or $P$). The maximum theoretical value of $RF_j$ is 1, which is achieved if site $j$ is simultaneously associated with the longest median lead time, the highest average seismic intensity, and the largest ambient population.

$RF_j$ is modified to account for alert accuracy (CA), as follows:

$$RF_{j,alert} = RF_j + F_{CA}(ca_j) \times w_{CA} \tag{4}$$

where $ca_j$ is the proportion of correct alerts at site $j$, calculated according to:

$$ca_j = \frac{n_{ca,j}}{n_j} \tag{5}$$

$n_j$ is the total number of catalogue earthquakes examined for $j$ (see "Seismic sources" section), which is all events from sources considered in the lead-time calculation that yield a predicted median PGA at the site of at least 0.001 g and result in either a false alert, a missed alert, or a correct alert ($n_{ca,j}$)—we ignore cases where the system correctly issues no alert, in line with Minson et al.[19]—and $w_P + w_I + w_L + w_{CA} = 1$. A false alert occurs if the predicted median PGA exceeds the threshold and the actual (randomly simulated) PGA does not, while a missed alert occurs in the opposite case. All other considered combinations of predicted median and actual ground shaking produce a correct alert.

### Data availability

The seismic station location data are available from the IRIS Google map (GMAP) station mapping service (http://ds.iris.edu/gmap/). The seismic source data and general target site locations are from the 2013 European Seismic Hazard Model[52,53]. The estimates of shear wave velocity in the uppermost 30 m are available in the database of Wald and Allen[57]. The velocity profiles are provided in the CRUST 1.0 dataset[61]. The population data are available in the 2018 Landscan database[70].

## Code availability

Code used in this research is available on Github at: https://github.com/gcrem/EEW_LEADTIME_EUROPE.

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

## Acknowledgements

We thank Dr. Alireza Azarbakht and Dr. John Douglas (University of Strathclyde, UK) for helpful feedback on parts of this study. All authors are supported by the European Union's Horizon 2020 research and innovation programme under grant agreement No. 821046, project TURNkey (Towards more Earthquake-resilient Urban Societies through a Multi-sensor-based Information System enabling Earthquake Forecasting, Early Warning and Rapid Response actions; https://earthquake-turnkey.eu/).

## Author contributions

G.C. and C.G. conceived and designed the research. G.C. drafted the written content of the manuscript, which all authors reviewed. G.C. performed the calculations. G.C. and C.G. developed the figures. E.Z. provided EEW and seismic hazard expertise.

## Competing interests

The authors declare no competing interests.
