## [Peer Review File · Nature Communications]

REVIEWER COMMENTS

Reviewer #1 (Remarks to the Author):

The paper examines Earthquake Early Warning feasibility at European level through a harmonized approach that involves lead-time computation (time available for security actions, while a seismic event is still developing on the fault), ground shaking intensity and exposed population. The authors summarize the information in a novel index, referred to as the Earthquake Early Warning relative feasibility index. As a result of their analysis, the authors claim that for many of the most hazardous regions in Europe, Earthquake Early Warning is a useful risk reduction tool, providing positive lead times for many of the analyzed scenarios. Although very interesting from an application point of view, the approach is grounded on too simplistic hypotheses (partly recognized also by the authors), that I believe strongly affect the results. Thus, I am not convinced of their validity. Also, the application of a complex algorithm for lead-time computation based on the numerical solution of the traveltimes eikonal equation is not a novel approach for Earthquake Early Warning. I detail my comments hereinafter.

Seismic stations

There are several issues with seismic stations. First the survey seems incomplete. Stations are taken from the IRIS catalog, that relies on the FDSN station service (https://www.fdsn.org/station_book/). This book contains information about stations from all networks that share data through the FDSN service, thus many stations but not all the stations. The problem is that if an institution in a country does not comply with the FDSN standard and thus does not share their own stations, the number of stations is completely wrong for that country and the following computation unreliable. From Figure 1a, we can recognize several countries with a very small number of stations because of this issue. For instance, the authors report that in Central and Southern Spain there are very few stations, while there are much more stations than those considered here (see for instance the ign web page <http://www.ign.es/web/recursos/sismologia/estaciones/estaciones.html>)

Also, I'm wondering if the authors restricted the selection to strong motion stations or considered all the available stations. In the station selection tool at IRIS I did not find the possibility to isolate strong-motion stations. Also, there is no mention of it in the paper. However, broad-band and short period stations are not suitable to record the strong-motion since they usually saturate, when located in the vicinity of the source. This may thus introduce an additional bias. In addition, there is no mention of the data telemetry. Are the stations analyzed here transmitting data in real-time? Also, there are several issues that affect the latency, for which we cannot simplify the problem to a constant contribution (smaller than 4s!). Many networks use the seedlink protocol, forcing the filling of the packet before sending it out. This can introduce a minimum latency up to about 3s, depending on the sampling rate. Thus, the consideration of a constant latency, hidden in the 4s of "average delay" does not represent a good estimate for all the stations in Europe and this value could be much larger for many actual operating networks. Since existing networks show a large latency and they need to be updated to serve for Earthquake Early Warning (EEW) purposes, I do not see the rationale in considering actual station distribution for EEW applications.

Source Models

As far as I understand the model is based on the hypothesis that events are point sources (fig. 2a). Also, the maximum magnitude in the catalog approaches M 8. An event of magnitude M 7 may rupture an area of 30-40 km length, while a rupture size of a M 8 event reaches a length of more than 100 km. Thus, constraining these events on single points and compute times from this model is too rough. Usually the PGA or other ground shaking indicators can be due to slip asperities far away from the hypocenter, significantly affecting the lead-time distribution and the results of Figure 3. It's worth to note that modern Early Warning systems are struggling to overcome this approximation since a while (see e.g. Böse, M., C. Felizardo, and T.H. Heaton, 2015: Finite-Fault Rupture Detector (FinDer): Going Real-Time in Californian ShakeAlert Warning System, *Seismol. Res. Lett.* 86(6), 1692-1704, doi:10.1785/0220150154)

Lead Times

The authors claim to overcome simple computations of lead time using a very complex physics-based travel-time computation. First, even simplistic methods use physics-based lead-times, they only limit the computation to a homogenous medium. However, computation of lead-time in more realistic media, using the Podvin and Lecomte algorithm, or NonLinLoc software is not a novel approach for EEW. It has been used since ten years (e.g. Satriano et al., 2011, PRESTo, the

earthquake early warning system for Southern Italy: Concepts, capabilities and future). Although the authors use an advanced tool for the travel time computation, they maintain a 1D model, which does not contain information about lateral variations in the crustal structure beneath Europe (e.g. Molinari and Morelli, 2011, EPcrust: a reference crustal model for the European Plate). This is a limitation. In addition, is there a real need of such a general approach for a 1D medium?

Relative Feasibility Index

Combination of lead time, intensity and population in a single index is a novel approach to evaluate the feasibility of an EEW system. However, Formula 4 is not clear because it sums up probability functions; which is the hypothesis beyond this sum? Also, the relation between the median values of the indicators and the probabilistic approach described in the methods section is not clear to me.

Uniform delay for computation

Beyond the telemetry latency, there is another important issue in EEW systems which is related to the computation time for obtaining an accurate estimate of the source parameters. Several studies indicate a saturation of the curves related to indicators that predict the magnitude. Saturation in magnitude depends on the P wave time window and may affect events with $M > 6.5$ for 3s of P wave time windows to $M > 7.5$ for 10s of P wave time window. There is a huge discussion in the seismological community on this topic. (see e.g. Trugman et al., 2019, Peak Ground Displacement Saturates Exactly When Expected: Implications for Earthquake Early Warning and references therein). Thus, a diverse "physical time" is required to perform an accurate estimation of the earthquake size as a function of the magnitude. Second, since the authors consider stations very close to the event epicenter, there is a contamination effect from S waves that strongly affect the time to get reliable estimations of the event size.

Reviewer #3 (Remarks to the Author):

In this study, the authors present a probabilistic evaluation of the usefulness of earthquake early warning in Europe, focusing on the Euro-Mediterranean region with some mentions of Iceland. Starting from the 2013 European Seismic Hazard model, they develop a novel measure to quantify the usefulness of EEW for a particular target site based on potential lead times, population density, and the ground acceleration for which a warning would be possible.

In the following I list major points that I think need to be addressed before publication. I have added further, minor comments to the manuscript itself.

1. In my opinion the authors overestimate the available lead time. One aspect of this is the inclusion of temporary seismic stations in the network analysis and the assumption that telemetry delays do not exceed 1-2 s. These are quite idealized conditions [1] and I think this should be made clearer in the main text. It is briefly mentioned in the discussion but I think it deserves more attention. Getting a seismic network EEW ready is a long and costly undertaking.

Further, in their lead time calculation, the authors assume that the final magnitude can be estimated from the first 1-2 s of the P-wave. With an additional 1-2 s for data telemetry delays and processing time, they assume that the final magnitude is known after a 4 s interval once a third station has detected the P-wave. Several studies have shown, however, that magnitude estimates from EEW algorithms using only the first few seconds of the P-wave recording tend to saturate at magnitudes greater than $M_w 6.5$ [2,3]. The likely reason is that rupture onset between an earthquake with $M_w \leq 6.5$ and $M_w > 6.5$ is indistinguishable during the first 4 seconds of the rupture [4]. For an $M_w 8$ it may even take up to 20 s after the rupture initiation before an EEW system could estimate the magnitude accurately [5]. As a consequence, targets further away from the epicenter do not necessarily have longer lead times [5]. I think this is an important aspect that the authors could include in their lead time calculations without too much additional work.

2. I am not completely convinced by the merit of the Relative Feasibility Index with the focus here on "Relative". If I understand correctly, in the hypothetical case that the maximum median lead time across all target sites was 2 s, a site with a median lead time of 1.9 s would have a high score even though the lead time would be too short for most mitigation actions. I think a function expressing the actual usefulness of an EEW alert would be more informative than the empirical cumulative distribution function.

3. I think the authors spend too much time in the discussion on reiterating their results and I missed a discussion on what would be necessary to improve EEW capability in the Euro-Mediterranean region.

Overall, the manuscript is well written and contains appropriate and balanced references to previous work. I believe it would be possible to reproduce results given the information in the manuscript. In my opinion, it is an important contribution to an ongoing discussion about the usefulness of earthquake early warning and contains some novel approaches to help answer this question.

Yannik Behr

References

1. Behr, Y. et al. The Virtual Seismologist in SeisComp3: A New Implementation Strategy for Earthquake Early Warning Algorithms. *Seismological Research Letters* 87, (2016).
2. Psimoulis, P. A., Houlié, N. & Behr, Y. Real-Time Magnitude Characterization of Large Earthquakes Using the Predominant Period Derived From 1 Hz GPS Data. *Geophysical Research Letters* 45, 517–526 (2018).
3. Hoshiya, M. & Iwakiri, K. Initial 30 seconds of the 2011 off the Pacific coast Tohoku earthquake ($M_w 9.0$) – amplitude and τ_c for magnitude estimation for earthquake early warning -. *Earth Planets Space* 63, 553–557 (2011).
4. Meier, M.-A., Heaton, T. & Clinton, J. F. Evidence for Universal Earthquake Rupture Initiation Behavior. 7991–7996 (2016) doi:10.1002/2016GL070081.
5. Minson, S. E., Meier, M.-A., Baltay, A. S., Hanks, T. C. & Cochran, E. S. The limits of earthquake early warning: Timeliness of ground motion estimates. *Science Advances* 4, eaaq0504 (2018).

Reviewer #2 (Remarks to the Author):

Disclaimer: I hold what seems to be a rather unconventional view on EEW. My recent paper (Wald, 2020) aimed to add some balance to the unmentioned and significant challenges of EEW that are not fully articulated by EEW proponents. *So, weight my perspectives as you see fit.*

This paper tackles an important question: Is EEW in Europe worth it? Will it help mitigate risk due to earthquake shaking? This study does so by creatively evaluating some the main issues: What are the distributions of EEW warning times possible in the Euro-Mediterranean region given the current seismic-station distribution, accurately estimated travel times, and probabilistic earthquake sources? With assumptions on the magnitude range of causative earthquakes that would potentially be mitigated with EEW, seismicity is used to evaluate warning times for different levels of shaking of societal relevance. With these lead times in hand, the evaluation of risk reduction is described by association with macroseismic intensity levels.

However, EEW proponents seem to be able to make optimistic claims, seemingly unchecked. I'm concerned that this work aims to prove a foregone conclusion rather than objectively testing a working scientific hypothesis. According to this analysis, the answer is clearly "yes" (EEW could mitigate risks across Europe), yet the evidence for that conclusion is less that satisfactory and the risk reduction *per se* is not actually quantified. Below, I provide what I consider to be a reality check on the "risk" analysis part of this study: Some of the basic assumption made—particularly the minimum magnitude 6.5 chosen and the low shaking thresholds selected for analyses (2% and 5%g)—result in overly optimistic warning times and the inferred potential for risk reduction.

My main overall concerns are followed by a number of additional comments. All said though, I am convinced of the importance of this topic and much of the analysis presented. So, I believe it should be published if the following concerns can be substantively addressed, ameliorated, or rebutted.

Main Concerns.

1. Perspective. It seems the authors are out to prove the utility of EEW for Europe as opposed to evaluate EEW objectively. First, the authors are involved with the project, so objectivity is questionable. I'm concerned when I see scientists—whose primary role is to seek out data or models to find evidence of the truth—take an active advocacy role rather than evaluate the available evidence. In contrast, Wald (2020) reviewed EEW potential for risk reduction, but he is *not* involved with EEW research and development. When advocacy is a primary component of a study, it may warrant an Opinion piece. For instance, there are no mentions or considerations of the numerous problems all fully operational EEW have experienced (M9 in Japan, M7 in Mexico, numerous false alerts and missed alerts in CA) as a real source of false/missed alerts. Assuming that perfect operational systems exist for the current study provide a rosy picture of what may be the reality. At least mention that this study assumes perfect systems. Many of the points below comeback to what I consider overly optimistic perspectives and assumptions without sufficient balance for the alternatives.
2. Definition of "warning time" and "lead time". "*In particular, we focus on EEW warning time (i.e., the time between the delivery of an EEW alert and the arrival of strong shaking at target sites)*".

Unlike Wald (2020), there is no mention of nor consideration of two critical aspects of any EEW system: (i) the time to deliver and communicate the warning to the users, and (2) the time the users take to cognate the message and take action. It borders on disingenuous to discuss risk mitigation specifically, without consideration that people and machines must receive, cognate, and take action. So, please define "warning time" and "lead time" carefully and note what they do and do not consider. Wald (2020) details how ignoring these additional latencies is misleading for users and for

any such benefit cost analyses. These additional latencies, of course, significantly reduce opportunities and options for EEW risk reduction.

3. Magnitudes considered. The use of events only greater than magnitude 6.5 distorts the statistics of exposure to intensity, again in the direction of favoring EEW. Wald (2020) note that for the fatal M5.2 earthquake in Osaka, Japan—in the middle of the world’s most developed EEW system—those in damaged areas where casualties occurred, were all within the blind zone. Many events in Europe well below M6.5 have been fatal, and for these, EEW is not likely to be beneficial. This is because the area with strong enough shaking to cause casualties is nearly entirely in the blind zone of even the best EEW systems. Such events are not factored into the accounting here despite their greater frequency.
4. Offshore earthquakes. It seems questionable to ignore hazards from offshore events. Would not the 1908 Messina Straits, Italy or the Great Lisbon earthquake be of concern and affect the statistics of this study? In this case, offshore events would actually benefit the average warning times, no?
5. Risk mitigation per se is not addressed in this study. The title “*Earthquake early warning could mitigate seismic risk across Europe*”, emphasizes risk mitigation. Yet, there are several issues not tackled that are essential to evaluating risk (and thus, evaluating if EEW might reduce risk):
 - a. “*We then determine the risk-reduction potential of these times, by defining their spatial relationship with exposure, event-dependent vulnerability, and an alert accuracy proxy, using well-established risk-prediction tools from earthquake engineering*” and “*Seismic intensity, which describes the effect of earthquake ground shaking on communities and the built environment, is a well-established proxy for both damage and vulnerability. We use the EMS-98 seismic intensity scale, which is specifically designed for European countries.*” Seismic intensity is most definitely not a proxy for damage. Intensities are not risk: only relating people to structures with specified vulnerabilities, shaken at specific intensities, can be loss or risk assessed. Combining population with intensity is not sufficient for addressing losses (or risk) without connecting them through a damage matrix specific to structures or a vulnerability function that provides a loss ratio, fatality, or injury rate. Risk involves **hazard, exposure, and vulnerability**. Using EMS-98 may indicate what effects may occur given specific structure types, but it is not a proxy for vulnerability. The average vulnerability in Switzerland is *not* the same as that in Italy, nor is Zurich the same as, say, Lugano. A building inventory (or the population’s vulnerability) is required to make this connection. Effectively, then, what you are saying about risk is really only a statement of the hazard.
 - b. Since this paper is primarily about alert time (not warning times; point above). Quantification of actual risk reduction was not adequately presented, “risk” probably should not be in the Title.
 - c. Discussion. “*We further contextualised the significance of the lead times by combining them with spatial distributions of two risk proxies often used in earthquake engineering (i.e., population and seismic intensity).*” Again, one cannot relate population and intensity to risk without knowing/specifying/estimating vulnerability and inventory.
 - d. “*... during this [warning] time, actions can be taken to significantly decrease detrimental impacts*”. This is stated as if fact, yet no examples have been provided of evidence of *decreased detrimental impacts*. Aren’t there examples given all the existing EEW systems mentioned? It is not defensible to make a general statement without citation to such examples? Simply citing others who make the same assertion in the EEW literature is not satisfactory. Perhaps the “*significantly decrease detrimental impacts*” part can be qualified or specified to make this statement scientifically accurate (*potentially, possibly, might*).
 - e. There is almost no discussion about the connection of the hazard level (PGA and intensity) and what actions may be taken at each level that will reduce losses at those levels.
 - f. All the above lines of reasoning, the caption for Figure 7 must be revised. (“*Examining the risk-mitigation potential of calculated lead times. Average EMS-98 macroseismic intensities experienced by the affected ambient population during events with a recurrence interval of 500 years that resulted in at least 0.05 g PGA at the associated target site, categorised by the*

corresponding times of the maps presented in Figure 4. Note that seismic intensities V, VI, and VII denote “strong”, “slightly damaging”, and “damaging” events, respectively”). Again, this has no bearing on risk. The *exposure* of a person to a given shaking has no bearing on their risk: They could be outside, in a car, in a safe structure, and at shaking levels as low as 2 or 5%g they are not risk in nearly any situation (See below).

6. EEW Feasibility Modelling. The so-called “relative feasibility index measure” can take on almost any values depending on the weighting and the choices of population and intensity (PGA in this case) parameters chosen. I’m not sure how valuable this metric is either in an absolute or a relative sense. The authors note that “*The indices of some target sites were significantly reduced (by up to almost 90%) in this case. However, this outcome is highly dependent on the assumed alert threshold (i.e., 0.05 g)*”²³. Using an alternative threshold of 0.02 g, for example, would have increased the accuracy of the worst-performing source-site combination by more than 20 times, due to less missed alerts.” The answer to the question of this study (EEW feasibility) is completely dependent on the choice of this weighting, and more specifically, on the threshold of PGA chosen. This means any outcome is possible and can be dialed-in based on the assumptions.
7. Intro. “*For each of these area sources, we calculate potential lead times (i.e., warning times that would be provided by EEW before shaking occurs) at target sites where the predicted median peak ground acceleration (PGA) associated with 500-year recurrence-interval events exceeds 0.05 g (see Figure 2b and Methods section), which is a commonly used threshold value for strong earthquake shaking in several engineering applications, including seismic design aimed at life-safety performance*”^{28–30}. You consider 5%g “strong shaking”, ShakeMap considers this “moderate” (Worden et al., 2020, figure below). First, if 5%g is strong, there are few adjectives left for 10, 20, 40 and 80%g? More to the point, selecting such a low threshold (5%g) for risk-reducing shaking makes EEW appear to be much more useful than it would be if you considered more reasonable (higher) shaking levels where risks are more obvious, but for which warning times are much shorter (e.g., Minson et al, 2019).

Moreover, a shaking value of 0.02g (low end of intensity IV) is too low for any risk considerations, as is 0.05g (low end intensity V). See the legend below from ShakeMap (Worden et al., 2020). If one chose a higher threshold of shaking, where damage actually occurs, then the results of this study would be much less favorable in terms of average alert times. Note: Minson (2019) also studied thresholds of EEW and considered four PGA levels a user might want to receive warning: 2%g, 5%g, 10%g, and 20%g. These correspond roughly to intensity levels IV (light shaking) through VII (very strong shaking). Note further that if life safety is a consideration for EEW, then one must again double the PGA level to 40%g for intensity VIII, where, by definition structural collapse *begins* and fatalities are almost always associated with collapse of buildings. In Japan, JMA warns the population for JMA intensity 5L for its public warnings. Intensity 5L is *at least* I_{MMI} VI, or closer to (I_{MMI} VII) depending on the conversion scale used, so somewhere around 10%g to 20%g.

Additional Comments.

- 1) Abstract: “*to an earthquake’s nucleation seconds to minutes before strong shaking occurs at target sites.*” “Minutes” (plural!) before is potentially applicable only to distant events; since you do not consider offshore events, the lead times you present can’t credibly include “*minutes*”, unless you are considering shaking levels well below societal relevance. This statement affords the critical reader to be immediately skeptical of what is to follow.
- 2) Introduction. Reference 2 (UNISDR. 2009) refers to early warning as in rapid alerting; this is not the same usage of the term *early warning* as used in *earthquake early warning*. In general, international multi-hazard agencies refer to quick reporting and impact assessment as “*early warning*”.
- 3) Intro. “*The process of EEW typically involves up to five main steps: (1) Detecting an earthquake, (2) estimating its location; (3) estimating its magnitude; (4) estimating the ground-motion at target sites; and (5) using all of the information collected to decide whether (or not) to trigger an alarm. EEW*

systems may be broadly categorised as “regional”, “on-site” (or “hybrid”) depending on their approach to the first three steps mentioned above.” Strictly speaking, on-site EEW systems do not actually need step 3: They typically predict the S wave amplitude based on P wave arrival (amplitude and period).

- 4) Intro. Ref #10, I don’t believe that Crowley et al. are specifically describing EEW as their strategy for risk reduction.
- 5) Intro. “*this is underlined by the fact that the average annual European GDP affected by earthquakes exceeds \$20 billion¹*”. RE: Reference #11. Please explain how EEW is to reduce any financial (losses) risk as is implicit in your statement. It has not (yet at least) been shown to be true. Or, if so, provide specific examples.
- 6) Intro. “*This study therefore offers a unique trans-national perspective on the potential of EEW that is highly relevant for intergovernmental stakeholders, such as the International Search and Rescue Advisory Group (INSARAG) of the United Nations²⁷*.” I’ve briefed INSARAG USAR teams numerous times. I find it questionable—or at least quite premature—to suggest that INSARAG is a EEW “stakeholder”. If I am wrong about this then please provide specifics.
- 7) Using the notion of “*minimum lead time*” is a bit misleading. Any region affected by crustal earthquakes has blind zones for all existing EEW systems, particularly for the damaging moderate-sized earthquake events.
- 8) Benchmarking the Calculations. “*The lead-time maps presented in Figure 4 are computed using a complex travel-time algorithm explicitly accounting for the physics of seismic wave propagation*. The terms “*complex*” and “*physics of seismic wave propagation*” seem a bit pretentious. Granted, you did not simply use a half-space approximation for travel times (as most do), but the calculations done are standard in other types of seismological analyses.
- 9) Benchmarking the Calculations. “*5% of minimum lead times, 16% of median lead times, and 64% of maximum lead times are over-estimated by at least one second when the simplified travel-time model is used, whereas 13% of minimum lead times, 8% of median lead times, and 4% of maximum lead times are under-estimated by at least the same amount*.” Given the generalized velocity structure employed, the use of “*complex, physics-based travel times*” is perhaps unnecessary. Other uncertainties in the calculation that were not considered are greater, for instance the evolution of rupture for $M > 6.5$ events, requiring additional time to properly quantify; and the fact that the initial P-waves and initial S-waves are not necessarily the pertinent arrivals. Likewise, the uncertainty in the delivery time is much greater, and that value has not even been considered.
- 10) “*we introduce a novel feasibility metric that enables identification of priority regions*”. As an Editor myself, I find the use of terms like “*novel*” and “*innovative*” presumptuous on the part of authors. Judgement of *novelty* and *innovative* should be rendered by the reader.
- 11) Risk Quantification. “*98% of the total ambient population*”. Here and elsewhere in this paragraph, one should avoid beginning a sentence with a numerical value.
- 12) Risk Quantification. How could higher intensities have longer lead times (higher percentage of pop. has > 10 sec of warning)? This conflicts with Minson's, Adrian-Meyer's, and all other studies, no?
- 13) Discussion. “*For example, they show that almost half (i.e., 44%) of the examined target sites benefit from warning times in a “best case scenario” that are long enough to accommodate major risk intervention actions, such as the shutting down of industrial equipment or the removal of vehicles from garages*”. First “*could (or might) benefit*” is more appropriate. Second, “*removal of vehicles from garages*” is not an action that I have ever come across as a potential benefit of EEW. Seems not so credible an action to me. Please provide a reference. Or, if you referring to opening garage doors at firehouses, that’s more specific that the wording you chose?
- 14) It would be nice to have a sense for why Iceland warrants EEW according to this study? It would be useful to attempt to explain how Iceland came to the top of the list? Tectonics? What are the controlling factors that make other countries more or less suitable for EEW?
- 15) Discussion. “*We did not explicitly consider the performance of existing EEW algorithms in the lead-time calculations. Instead, we simply assumed that four seconds was sufficient to capture both data*”

telemetry delays and the length of time required by the algorithms to estimate relevant source parameters.” This is a rather essential assumption that affects the results. This should be stated much earlier in the manuscript.

- 16) Competing interests. “The authors declare no competing interests”. I’m not sure I completely agree with this declaration. The authors state note:

“A new three-year Horizon 2020 European research project called TURNkey (Towards more Earthquake-resilient Urban Societies through a Multi-sensor-based Information System enabling Earthquake Forecasting, Early Warning and Rapid Response actions; see Acknowledgements section) aims to address this issue by developing a holistic earthquake information system that incorporates state-of-the-art seismic risk mitigation tools for both operational earthquake forecasting and EEW in real- and near-real time, with selected testbeds in Italy and Greece (as well as Iceland) to be the focus of more detailed analyses for EEW and the target of end-user-orientated applications of the system.”

The Acknowledgements note that this study was funded by Horizon 2020. The statement quoted above is quite clearly advocacy for Horizon2020/Turnkey. Is that *not* a competing interest?

References:

Minson SE, Baltay AS, Cochran ES, Hanks TC, Page MT, McBride SK, Milner KR, Meier M-A (2019): The limits of earthquake early warning accuracy and best alerting strategy. *Scientific Reports*, **9**(1), 2478.

Wald, David J. “Practical Limitations of Earthquake Early Warning.” *Earthquake Spectra* 36, no. 3, 1412–47. <https://doi.org/10.1177/8755293020911388>.

Worden CB, Thompson EM, Hearne MG, Wald DJ (2020) ShakeMap V4 manual: technical manual user’s guide, and software guide. <http://usgs.github.io/shakemap>

ShakeMap Legend and approximate intensity conversion scale (Worden et al, 2020).

SHAKING	Not felt	Weak	Light	Moderate	Strong	Very strong	Severe	Violent	Extreme
DAMAGE	None	None	None	Very light	Light	Moderate	Moderate/heavy	Heavy	Very heavy
PGA(%g)	<0.05	0.3	2.76	6.2	11.5	21.5	40.1	74.7	>139
PGV(cm/s)	<0.02	0.13	1.41	4.65	9.64	20	41.4	85.8	>178
INTENSITY	I	II-III	IV	V	VI	VII	VIII	IX	X+

Response to comments for “Earthquake early warning could mitigate seismic risk across Europe”, by Gemma Cremen, Carmine Galasso, and Elisa Zuccolo

NCOMMS-20-42346

We thank the reviewers for their thoughtful and insightful comments, which have significantly improved the quality of the revised manuscript. The reviewer’s comments have been numbered and listed below (in red), followed by our responses in italics.

Reviewer #1:

The paper examines Earthquake Early Warning feasibility at European level through a harmonized approach that involves lead-time computation (time available for security actions, while a seismic event is still developing on the fault), ground shaking intensity and exposed population. The authors summarize the information in a novel index, referred to as the Earthquake Early Warning relative feasibility index. As a result of their analysis, the authors claim that for many of the most hazardous regions in Europe, Earthquake Early Warning is a useful risk reduction tool, providing positive lead times for many of the analyzed scenarios. Although very interesting from an application point of view, the approach is grounded on too simplistic hypotheses (partly recognized also by the authors), that I believe strongly affect the results. Thus, I am not convinced of their validity. Also, the application of a complex algorithm for lead-time computation based on the numerical solution of the travel-time eikonal equation is not a novel approach for Earthquake Early Warning. I detail my comments hereinafter.

We highly value the reviewer’s comments and have made extensive changes to the content of the manuscript (described in detail in the comments below) in response. In particular, we have made significant modifications to the design of the study that more realistically reflect the (often limiting) practicalities of operational EEW. We agree that the travel time algorithm used in our study has been used in previous earthquake early warning studies and, as such, we don’t claim its use as a novel point of our study. We apologise if this was the impression provided to a reader of the original manuscript. As stated by the reviewer, the earthquake early warning relative feasibility index is the main novelty of the study, and the results of using the aforementioned travel time algorithm comprise just one ingredient of this index. In addition, we note that this study represents the first attempt to use this type of travel time algorithm for EEW on a continent-wide scale – in conjunction with a number of seismicity scenarios as well as other assumptions/models – to compute probabilistic distributions of lead times across Europe.

1. Seismic stations

There are several issues with seismic stations. First the survey seems incomplete. Stations are taken from the IRIS catalog, that relies on the FDSN station service (https://www.fdsn.org/station_book/). This book contains information about stations from all networks that share data through the FDSN service, thus many stations but not all the stations. The problem is that if an institution in a country does not comply with the FDSN standard and thus does not share their own stations, the number of stations is completely wrong for that country and the following computation unreliable. From Figure 1a, we can recognize several countries with a very small number of stations because of this issue. For instance, the authors report that in Central and Southern Spain there are very few stations, while there are much more stations than those considered here (see for instance the ign web page <http://www.ign.es/web/recursos/sismologia/estaciones/estaciones.html>)

Thanks for this comment. We recognise that using the IRIS catalog may not capture all stations across Europe, and therefore that our calculations may be conservative (i.e., producing an underestimation of actual lead times) in some cases. However, we have opted to keep this data source for the following reasons:

1. We believe it is important to use a consistent data source across the entire study region.
2. Some countries may not provide access to the locations of all their seismic stations (e.g., Malta – see “Data and Resources” section of [1]); thus, an exhaustive search of each country’s seismic station catalog may not necessarily substantially increase the reliability (or reduce the bias) of the computation.
3. Many previous studies on European seismology have also exclusively relied on seismic station data from international databases (e.g., IRIS, EIDA) [2-4].

References:

- [1] Galea, P., Bozionelos, G., D’Amico, S., Drago, A., & Colica, E. (2018). Seismic Signature of the Azure Window Collapse, Gozo, Central Mediterranean. *Seismological Research Letters*, 89(3), 1108-1117.
- [2] Yang, Y., Ritzwoller, M. H., Levshin, A. L., & Shapiro, N. M. (2007). Ambient noise Rayleigh wave tomography across Europe. *Geophysical Journal International*, 168(1), 259-274.
- [3] Retailleau, L., Boué, P., Li, L., & Campillo, M. (2020). Ambient seismic noise imaging of the lowermost mantle beneath the North Atlantic Ocean. *Geophysical Journal International*, 222(2), 1339-1351.
- [4] Blom, N., Gokhberg, A., & Fichtner, A. (2020). Seismic waveform tomography of the central and eastern Mediterranean upper mantle. *Solid Earth*, 11(2), 669-690.

We have added the following text in the Discussion section, to acknowledge the potential limitations of using an international seismic station database: “.. we leveraged an international database to obtain details on seismic stations (see Methods section). While this approach ensured a consistent data source was used across the entire study region, it may not have completely captured all stations across Europe. Thus, our calculations may have underestimated actual lead times in some cases.”

Also, I’m wondering if the authors restricted the selection to strong motion stations or considered all the available stations. In the station selection tool at IRIS I did not find the possibility to isolate strong-motion stations. Also, there is no mention of it in the paper. However, broadband and short period stations are not suitable to record the strong-motion since they usually saturate, when located in the vicinity of the source. This may thus introduce an additional bias.

We considered all available stations in the original study. However, we acknowledge that this was not the most appropriate approach for an earthquake early warning study. We have therefore removed the following stations from our revised analyses:

- all temporary stations
- all permanent stations that are not strong-motion or broadband. Note that broadband stations are selected from IRIS using the channel search queries “HH?” and “BH?”, and strong-motion stations are selected from IRIS using the channel search query “?N?”.

We note that the reviewer does not think that broadband stations are suitable for our calculations. However, we have chosen to include them in our revised analyses given that:

1. Earthquake early warning algorithms have been specifically designed based on their data [1] and
2. Broadband stations are used in practical applications of earthquake early warning, for example the ShakeAlert Earthquake Early Warning System in California [2].

We note that real-time systems usually apply a clipping threshold to prevent the use of saturated velocigrams; this is not possible in this study, given that actual waveforms are not used. In any case, we assume that the broadband sensors could be upgraded to strong-motion sensors that are capable of transmitting data in real time. This assumption is stated in the last paragraph of the Discussion section: “Secondly, it is assumed that the considered seismic stations are (or could be) capable of being used for

early warning purposes (i.e., they have or could have adequate data acquisition/transmission systems, real-time communication capability, robust dissemination methods, power supply systems, etc. [1,2])”.

References:

- [1] Galea, P., Bozionelos, G., D’Amico, S., Drago, A., & Colica, E. (2018). *Seismic Signature of the Azure Window Collapse, Gozo, Central Mediterranean*. *Seismological Research Letters*, 89(3), 1108-1117.
- [2] Chung, A. I., Meier, M. A., Andrews, J., Böse, M., Crowell, B. W., McGuire, J. J., & Smith, D. E. (2020). *ShakeAlert earthquake early warning system performance during the 2019 Ridgecrest earthquake sequence*. *Bulletin of the Seismological Society of America*, 110(4), 1904-1923.

The description of Seismic Station Data in the “Methods” section now reads: “We consider all permanent strong-motion and broadband stations between -26° and 45° longitude, and 34° and 72° latitude.”

In addition, there is no mention of the data telemetry. Are the stations analyzed here transmitting data in real-time? Also, there are several issues that affect the latency, for which we cannot simplify the problem to a constant contribution (smaller than 4s!). Many networks use the seedlink protocol, forcing the filling of the packet before sending it out. This can introduce a minimum latency up to about 3s, depending on the sampling rate. Thus, the consideration of a constant latency, hidden in the 4s of “average delay” does not represent a good estimate for all the stations in Europe and this value could be much larger for many actual operating networks.

Thanks for pointing this out. No, not all the stations considered here are transmitting data in real time, but we are assuming that they could be upgraded/replaced for use in an EEW system. This assumption is stated in the last paragraph of the Discussion section: “Secondly, it is assumed that the considered seismic stations are (or could be) capable of being used for early warning purposes (i.e., they have or could have adequate data acquisition/transmission systems, real-time communication capability, robust dissemination methods, power supply systems, etc. [1,2])”. We therefore assume data telemetry delay times that correspond with real-time communication capabilities (as opposed to current system configurations that may have larger latencies, as noted by the reviewer). These delays are now explicitly incorporated in the lead-time calculation (along with magnitude-dependent time windows required to compute characteristics of the ongoing event), and (in line with previous EEW studies [3-5]) are assumed to comprise of 1 second for data transmission and 1 second for issuing the warning message. We comment on the ideal nature of this delay in two places:

1. *“Lead-Time Mapping for High Hazard Areas” section now contains the following line: “These calculations incorporate a magnitude-dependent delay interval that captures the time required to compute characteristics of the ongoing event and to complete a state-of-the-art real-time data telemetry process (see Methods section).”*
2. *“Lead-time Modelling” in Methods “[The data telemetry time] captures data telemetry delays, which are idealistically assumed to comprise 1 second for data transmission and 1 second for issuing the warning message [3-5].”*

References:

- [1] Auclair, S., Goula, X., Jara, J. A., & Colom, Y. (2015). *Feasibility and interest in earthquake early warning systems for areas of moderate seismicity: Case study for the Pyrenees*. *Pure and Applied Geophysics*, 172(9), 2449-2465.
- [2] Thelen, W. A., Hotovec-Ellis, A. J., & Bodin, P. (2016). *Feasibility study of earthquake early warning (EEW) in Hawaii (No. 2016-1172)*. US Geological Survey.
- [3] Allen, R. M. (2007). *The ElarmS earthquake early warning methodology and application across California*. In *Earthquake early warning systems* (pp. 21-43). Springer, Berlin, Heidelberg.
- [4] Behr, Y., Clinton, J., Kästli, P., Cauzzi, C., Racine, R., & Meier, M. A. (2015). *Anatomy of an earthquake early warning (EEW) alert: Predicting time delays for an end-to-end EEW system*. *Seismological Research Letters*, 86(3), 830-840.

[5] Zuccolo, E., Gibbs, T., Lai, C. G., Latchman, J. L., Salazar, W., Di Sarno, L., ... & Workman, A. (2016). Earthquake early warning scenarios at critical facilities in the Eastern Caribbean. *Bulletin of Earthquake Engineering*, 14(9), 2579-2605.

Since existing networks show a large latency and they need to be updated to serve for Earthquake Early Warning (EEW) purposes, I do not see the rationale in considering actual station distribution for EEW applications.

While this is true, we nevertheless believe a reasonable first-principles approach to examining the potential effectiveness of earthquake early warning is to use current seismic station locations (i.e., accounting for the geometrical characteristics of the network and assuming that necessary hardware and software upgrades for EEW are possible). Note that this approach is directly in line with previous studies that have examined EEW feasibility [1,2].

We have added a note on this approach at the start of the “Seismic Stations” subsection of Methods, as follows: “We use current seismic station locations in this work (and thus account for the geometrical characteristics of the network, assuming that necessary hardware/software upgrades for EEW are possible), in line with previous studies that have examined EEW feasibility [1,2].”

References:

[1] Picozzi, M., Zollo, A., Brondi, P., Colombelli, S., Elia, L., & Martino, C. (2015). Exploring the feasibility of a nationwide earthquake early warning system in Italy. *Journal of Geophysical Research: Solid Earth*, 120(4), 2446-2465.

[2] Allen, R. M. (2006). Probabilistic warning times for earthquake ground shaking in the San Francisco Bay Area. *Seismological Research Letters*, 77(3), 371-376.

2. Source Models

As far as I understand the model is based on the hypothesis that events are point sources (fig. 2a). Also, the maximum magnitude in the catalog approaches M 8. An event of magnitude M 7 may rupture an area of 30-40 km length, while a rupture size of a M 8 event reaches a length of more than 100 km. Thus, constraining these events on single points and compute times from this model is too rough. Usually the PGA or other ground shaking indicators can be due to slip asperities far away from the hypocenter, significantly affecting the lead-time distribution and the results of Figure 3. It's worth to note that modern Early Warning systems are struggling to overcome this approximation since a while (see e.g. Böse, M., C. Felizardo, and T.H. Heaton, 2015: Finite-Fault Rupture Detector (FinDer): Going Real-Time in Californian ShakeAlert Warning System, *Seismol. Res. Lett.* 86(6), 1692-1704, doi:10.1785/0220150154)

We appreciate this comment. However, it is important to note that this study considers area zones discretized into 0.1°x0.1° cells and that these cells can represent point sources or asperity zones of extended faults. We constrain each event to one point (i.e., cell) for the lead-time calculations, in line with other authors like Picozzi et al. [1], Kuyuk and Allen [2], and Parolai et al. [3].

However, we have made the following changes to our calculations to more realistically capture the rupture of large magnitude earthquakes:

- *We now use Joyner-Boore distances rather than epicentral distances to compute PGA, to better account for fault dimensions. We obtain Joyner-Boore distances from epicentral distances, using the adjustment factors provided in [1] for the style-of-faulting and tectonic setting (i.e., active crustal or stable continental) of the associated seismic source.*

- Lead times are now computed for sites where median PGA predictions for the Joyner-Boore version of the [2] GMM exceed 0.05g.

References:

- [1] Thompson, E. M., & Worden, C. B. (2018). Estimating Rupture Distances without a Rupture. *Bulletin of the Seismological Society of America*, 108(1), 371-379.
- [2] Akkar, S., Sandikkaya, M. A., & Bommer, J. J. (2014). Empirical ground-motion models for point-and extended-source crustal earthquake scenarios in Europe and the Middle East. *Bulletin of earthquake engineering*, 12(1), 359-387.
- [3] Picozzi, M., Zollo, A., Brondi, P., Colombelli, S., Elia, L., & Martino, C. (2015). Exploring the feasibility of a nationwide earthquake early warning system in Italy. *Journal of Geophysical Research: Solid Earth*, 120(4), 2446-2465.
- [4] Kuyuk, H. S., & Allen, R. M. (2013). Optimal seismic network density for earthquake early warning: A case study from California. *Seismological Research Letters*, 84(6), 946-954.
- [5] Parolai, S., Boxberger, T., Pilz, M., Fleming, K., Haas, M., Pittore, M., ... & Lauterjung, J. (2017). Assessing earthquake early warning using sparse networks in developing countries: Case study of the Kyrgyz Republic. *Frontiers in Earth Science*, 5, 74.

3. Lead Times

The authors claim to overcome simple computations of lead time using a very complex physics-based travel-time computation. First, even simplistic methods use physics-based lead-times, they only limit the computation to a homogenous medium. However, computation of lead-time in more realistic media, using the Podvin and Lecomte algorithm, or NonLinLoc software is not a novel approach for EEW. It has been used since ten years (e.g. Satriano et al., 2011, PRESTo, the earthquake early warning system for Southern Italy: Concepts, capabilities and future). Although the authors use an advanced tool for the travel time computation, they maintain a 1D model, which does not contain information about lateral variations in the crustal structure beneath Europe (e.g. Molinari and Morelli, 2011, EPcrust: a reference crustal model for the European Plate). This is a limitation. In addition, is there a real need of such a general approach for a 1D medium?

Thanks for this comment. We agree that our description of this algorithm was overstated. To rectify this issue, all references to “sophisticated travel-time” and “complex travel-time” model/algorithm have been replaced with “grid-based finite-difference travel-time” model/algorithm, and all references to wave physics or wave propagation have been removed. We agree that using the NonLinLoc software for earthquake early warning is not novel. It is important to note, however, that this study represents the first attempt to use this type of travel time algorithm for EEW on a continent-wide scale – in conjunction with a number of seismicity scenarios as well as other assumptions/models – to compute probabilistic distributions of lead times across Europe.

We believe that a 1-D model is acceptable in this case, given that this type of model is routinely used in the NonLinLoc software for EEW purposes, i.e., the PRESTo platform (<http://www.prestoews.org/documentation.php>) mentioned by the reviewer.

Furthermore, we believe that notable discrepancies between the lead-time maps calculated using the NonLinLoc software and those obtained using the simplified lead-time calculation (that only considers the direct arrival of P-waves and a single average value of P- or S-wave velocity; see Figure 1) justify our physics-based approach to travel-time calculation, regardless of the 1D simplification.

However, we appreciate the reviewer’s opinion that use of a 1D model is a limitation of our analysis, and we have added the following note in the Discussion section to highlight this simplifying assumption: “We used a 1-D velocity model in the travel-time algorithm, which does not capture lateral variations in the earth’s structure.”

Figure 1: Differences obtained in lead-time maps, using the simplified travel-time model that only considers the direct arrival of P-waves and a single average value for both P- and S-wave velocity. (a) Minimum lead times; (b) median lead times; and (c) maximum lead times. Note that the maximum absolute discrepancy in each case is (a) 11.5; (b) 10.3; and (c) 6.6 seconds.

4. Relative Feasibility Index

Combination of lead time, intensity and population in a single index is a novel approach to evaluate the feasibility of an EEW system. However, Formula 4 is not clear because it sums up probability functions; which is the hypothesis beyond this sum?

The relative feasibility index is used to compare the feasibility (or “usefulness”) of EEW for target sites, in a relative sense. The empirical cumulative distribution functions are used to rank the various sites based on average seismic intensity, (positive) median lead time, and ambient population. The highest possible value of the index is 1, which would be achieved for a site that ranked first across all three metrics, i.e., a site that is associated with the highest average seismic intensity, the longest lead time, and the largest ambient population.

The index is explained in the main text as follows: “Higher values of this index correspond to key characteristics that maximise the effectiveness of an EEW system [1], i.e.: (1) longer lead times; (2) higher potential for shaking causing losses that can be avoided with EEW; and (3) larger affected populations. They, therefore, indicate greater EEW feasibility for a given target site.”

We have added the following text in the “Methods” section under “EEW Feasibility Modelling”, to better convey the meaning/interpretation of the index: “Each $F_X(.)$ function ranks the sites based on the underlying metric (i.e., L , I or P). The maximum theoretical value of RF_j is 1, which is achieved if site j is simultaneously associated with the longest median lead time, the highest average seismic intensity, and the largest ambient population.”

References:

[1] Kuyuk, H. S., & Allen, R. M. (2013). Optimal seismic network density for earthquake early warning: A case study from California. *Seismological Research Letters*, 84(6), 946-954.

Also, the relation between the median values of the indicators and the probabilistic approach described in the methods section is not clear to me.

The purpose of the probabilistic dimension of the index (i.e., the empirical cumulative distribution functions) is simply to rank the sites according to the various metrics considered. We rank the sites based on average or median values of these metrics. This is now clarified in the text, as described in our response to the previous point.

5. Uniform delay for computation

Beyond the telemetry latency, there is another important issue in EEW systems which is related to the computation time for obtaining an accurate estimate of the source parameters. Several studies indicate a saturation of the curves related to indicators that predict the magnitude. Saturation in magnitude depends on the P wave time window and may affect events with $M > 6.5$ for 3s of P wave time windows to $M > 7.5$ for 10s of P wave time window. There is a huge discussion in the seismological community on this topic. (see e.g. Trugman et al., 2019, Peak Ground Displacement Saturates Exactly When Expected: Implications for Earthquake Early Warning and references therein). Thus, a diverse “physical time” is required to perform an accurate estimation of the earthquake size as a function of the magnitude. Second, since the authors consider stations very close to the event epicenter, there is a contamination effect from S waves that strongly affect the time to get reliable estimations of the event size.

Thanks for this comment. To address it, we now incorporate a delay in our lead-time calculations that specifically accounts for the time required to compute different earthquake magnitudes. This delay time is assigned in line with [1]; it is assumed to equal 3 seconds for magnitudes less than 6.5, 4 seconds for magnitudes between 6.5 and 7, 12 seconds for magnitudes between 7 and 7.5, and 20 seconds for magnitudes greater than 7.5. The value of this delay used for a given area source is based on the magnitude of the event with a recurrence interval of 500 years (except when magnitude 5 events are examined, in which case δm is uniformly set to 3 seconds). We have added the following text on these delay times to “Lead-time Modelling” in Methods: δm represents the time required to compute the magnitude of the ongoing event, and is assumed to equal 3 seconds for $M_w < 6.5$, 4 seconds for $6.5 \leq M_w < 7$, 12 seconds for $7 \leq M_w < 7.5$, and 20 seconds for $M_w \geq 7.5$ [1]. The value of δm used for a given area source is based on the magnitude of the event with a recurrence interval of 500 years (except when M_w 5 events are examined, in which case δm is uniformly set to 3 seconds). Note that this delay is added to the 2-second data telemetry time (see our response to Comment #1 for more details) to determine the total delay interval for a given lead-time calculation.

References:

[1] Trugman, D. T., Page, M. T., Minson, S. E., & Cochran, E. S. (2019). Peak ground displacement saturates exactly when expected: Implications for earthquake early warning. *Journal of Geophysical Research: Solid Earth*, 124(5), 4642-4653.

Figure 2 below shows the minimum, median, and maximum lead-time maps for the modified delay intervals considered in the updated manuscript. 3% of sites have positive minimum lead times, 18% of sites have positive median lead times, and 79% of sites have positive maximum lead time. Please see the “Lead-Time Mapping for High Hazard Areas” section of the manuscript for a more complete discussion of the results.

Figure 2: (a) Minimum, (b) Median, and (c) Maximum lead-time maps resulting from the updated lead-time calculations.

Reviewer #2 (Dr. David Wald):

Disclaimer: I hold what seems to be a rather unconventional view on EEW. My recent paper (Wald, 2020) aimed to add some balance to the unmentioned and significant challenges of EEW that are not fully articulated by EEW proponents. *So, weight my perspectives as you see fit.*

This paper tackles an important question: Is EEW in Europe worth it? Will it help mitigate risk due to earthquake shaking? This study does so by creatively evaluating some the main issues: What are the distributions of EEW warning times possible in the Euro-Mediterranean region given the current seismic-station distribution, accurately estimated travel times, and probabilistic earthquake sources? With assumptions on the magnitude range of causative earthquakes that would potentially be mitigated with EEW, seismicity is used to evaluate warning times for different levels of shaking of societal relevance. With these lead times in hand, the evaluation of risk reduction is described by association with macroseismic intensity levels.

However, EEW proponents seem to be able to make optimistic claims, seemingly unchecked. I'm concerned that this work aims to prove a foregone conclusion rather than objectively testing a working scientific hypothesis. According to this analysis, the answer is clearly "yes" (EEW could mitigate risks across Europe), yet the evidence for that conclusion is less that satisfactory and the risk reduction *per se* is not actually quantified. Below, I provide what I consider to be a reality check on the "risk" analysis part of this study: Some of the basic assumption made—particularly the minimum magnitude 6.5 chosen and the low shaking thresholds selected for analyses (2% and 5%g)—result in overly optimistic warning times and the inferred potential for risk reduction.

My main overall concerns are followed by a number of additional comments. All said though, I am convinced of the importance of this topic and much of the analysis presented. So, I believe it should be published if the following concerns can be substantively addressed, ameliorated, or rebutted.

We appreciate your comments and have made extensive changes to the content of the manuscript (described in detail in the comments below) in response. It is important to note that the aim of this study is to provide an objective assessment of the feasibility of EEW in Europe (rather than to advocate for its use), and we believe the significant changes we have made to the design of the study serve to better emphasise this motivation. Although the original manuscript highlighted the "positive" outcomes of the feasibility analysis, it did also contain some negative results (e.g., it was clear that EEW was not useful in many places throughout Europe); the revised manuscript presents both positive and negative aspects of EEW feasibility in a more balanced way.

1. Perspective. It seems the authors are out to prove the utility of EEW for Europe as opposed to evaluate EEW objectively. First, the authors are involved with the project, so objectivity is questionable. I'm concerned when I see scientists—whose primary role is to seek out data or models to find evidence of the truth—take an active advocacy role rather than evaluate the available evidence. In contrast, Wald (2020) reviewed EEW potential for risk reduction, but he is *not* involved with EEW research and development. When advocacy is a primary component of a study, it may warrant an Opinion piece. For instance, there are no mentions or considerations of the numerous problems all fully operational EEW have experienced (M9 in Japan, M7 in Mexico, numerous false alerts and missed alerts in CA) as a real source of false/missed alerts. Assuming that perfect operational systems exist for the current study provide a rosy picture of what may be the reality. At least mention that this study assumes perfect systems. Many of the points below comeback to what I consider overly optimistic perspectives and assumptions without sufficient balance for the alternatives.

*We agree that some of our assumptions in the original study were oversimplifications, which seem to have unfortunately provided the misleading impression that we "are out to prove the utility of EEW for Europe". Our intention is to provide an objective analysis of EEW feasibility in Europe (**note that the source of our funding does not – and would never - affect our objectivity in any way** – see later comment on this), and we have substantially*

modified/extended our calculations to better represent this motivation. In particular, these modifications/extensions are based on your points/suggestions below and include:

1. Examining EEW lead times for magnitude-dependent delay times reported in the literature, to account for longer latencies that may be necessary for determining the magnitudes of larger events, for example.
2. Also examining EEW lead times for smaller events (i.e., magnitude 5 earthquakes).
3. Also examining EEW lead times for higher limits on seismic intensity (i.e., only considering sites where the median PGA prediction exceeds 0.1g for the prescribed event).

These modifications/extensions to the calculations are discussed in more detail in following comments.

In addition, we have extensively re-phrased our language around the concept of “risk” (please see our response to your comment #6 for a more detailed explanation on the related modifications).

It is important to note that false and missed alert occurrences are considered in the feasibility study; in the second half of the Relative Feasibility Index calculations, the Relative Feasibility Index is modified to account for alert accuracy, explicitly considering the potential occurrence of missed or false warnings (in line with Minson et al [1]).

References:

[1] Minson, S. E., Baltay, A. S., Cochran, E. S., Hanks, T. C., Page, M. T., McBride, S. K., ... & Meier, M. A. (2019). The limits of earthquake early warning accuracy and best alerting strategy. *Scientific reports*, 9(1), 1-13.

2. Definition of “warning time” and “lead time”. “In particular, we focus on EEW warning time (i.e., the time between the delivery of an EEW alert and the arrival of strong shaking at target sites)”.

Unlike Wald (2020), there is no mention of nor consideration of two critical aspects of any EEW system: (i) the time to deliver and communicate the warning to the users, and (2) the time the users take to cognate the message and take action. It borders on disingenuous to discuss risk mitigation specifically, without consideration that people and machines must receive, cognate, and take action. So, please define “warning time” and “lead time” carefully and note what they do and do not consider. Wald (2020) details how ignoring these additional latencies is misleading for users and for any such benefit cost analyses. These additional latencies, of course, significantly reduce opportunities and options for EEW risk reduction.

Thanks for this comment. We acknowledge that use of the term “warning time” is misleading in this case, given that we are not accounting for the time required to receive and respond to the alert. We have therefore replaced all references to warning time that implied lead time with the term “lead time”. In addition, we have re-phrased the following text in the “Lead-Time Mapping for High Hazard Areas” section: “...we calculate potential lead times (i.e., warning times that would be provided by EEW before shaking occurs” as “...we calculate potential lead times (i.e., times between EEW alert issuance and the occurrence of shaking)”

In addition, we have added a cautionary note to our discussion of possible actions that can be taken for various lead times in the “Discussion” section, as follows: “The times considered in Table 1 for taking prescribed actions during an EEW alert may be longer in practical cases, given that human reaction latencies have not yet been well established in this context [1].”

References:

[1] Wald, D. J. (2020). Practical limitations of earthquake early warning. *Earthquake Spectra*, 36(3), 1412-1447.

3. Magnitudes considered. The use of events only greater than magnitude 6.5 distorts the statistics of exposure to intensity, again in the direction of favoring EEW. Wald (2020) note that for the fatal M5.2 earthquake in Osaka, Japan—in the middle of the world’s most developed EEW system—those

in damaged areas where casualties occurred, were all within the blind zone. Many events in Europe well below M6.5 have been fatal, and for these, EEW is not likely to be beneficial. This is because the area with strong enough shaking to cause casualties is nearly entirely in the blind zone of even the best EEW systems. Such events are not factored into the accounting here despite their greater frequency.

Thanks for this comment. To address it, we have added a section (“Lead-Time Sensitivity Analyses”) that specifically examines lead times for magnitude 5 earthquakes at sites where the associated median PGA exceeds 0.05 g. Figure 3 below shows the minimum, median, and maximum lead-time maps for this case. Note that only Figure 3c is displayed in the revised version of the manuscript (as Figure 5), given that the three maps are identical (i.e., all of the study area is within the blind zone for the three considered scenarios). Our comments on these calculations can be found in the “Lead-Time Sensitivity Analyses” section of the revised manuscript as follows: “No target sites have positive maximum lead times in this case.” The findings of these calculations contribute to the following conclusion made in the Discussion section: “... the viability of EEW is highly dependent on the magnitude of the ongoing event and the threshold PGA at which an EEW alert is issued at a target site.”

Figure 3: (a) Minimum, (b) Median, and (c) Maximum lead-time maps for magnitude 5 events.

4. Offshore earthquakes. It seems questionable to ignore hazards from offshore events. Would not the 1908 Messina Straits, Italy or the Great Lisbon earthquake be of concern and affect the statistics of this study? In this case, offshore events would actually benefit the average warning times, no?

We do consider offshore earthquakes. Figure 2(a) shows the earthquake sources considered in our study, some of which are offshore.

5. Risk mitigation per se is not addressed in this study. The title “Earthquake early warning could mitigate seismic risk across Europe”, emphasizes risk mitigation. Yet, there are several issues not tackled that are essential to evaluating risk (and thus, evaluating if EEW might reduce risk):

Thanks for this comment. We agree with you that risk mitigation is not explicitly addressed in this study, and we have made a number of modifications to the manuscript to clarify this issue. Please see the response to each sub-point below, for more details.

- a. “We then determine the risk-reduction potential of these times, by defining their spatial relationship with exposure, event-dependent vulnerability, and an alert accuracy proxy, using well-established risk-prediction tools from earthquake engineering” and “Seismic

intensity, which describes the effect of earthquake ground shaking on communities and the built environment, is a well-established proxy for both damage and vulnerability. We use the EMS-98 seismic intensity scale, which is specifically designed for European countries.” Seismic intensity is most definitely not a proxy for damage. Intensities are not risk: only relating people to structures with specified vulnerabilities, shaken at specific intensities, can be loss or risk assessed. Combining population with intensity is not sufficient for addressing losses (or risk) without connecting them through a damage matrix specific to structures or a vulnerability function that provides a loss ratio, fatality, or injury rate. Risk involves **hazard, exposure, and vulnerability**. Using EMS-98 may indicate what effects may occur given specific structure types, but it is not a proxy for vulnerability. The average vulnerability in Switzerland is *not* the same as that in Italy, nor is Zurich the same as, say, Lugano. A building inventory (or the population’s vulnerability) is required to make this connection. Effectively, then, what you are saying about risk is really only a statement of the hazard.

We agree that seismic intensity on its own is not a direct proxy of damage, and that the study does not deal with explicit measures of vulnerability. Therefore, we have removed direct references to the concept of risk, in the context of our study (see below for a list of changes made). Nevertheless, population is a proxy for exposure – e.g., it is used to approximate spatial building distributions in regional (seismic) risk analyses [1,2] - and therefore we do not agree that what we are saying about risk is only a statement about hazard.

To address this comment, we have made the following changes:

- *The following sentence highlighted by the reviewer: “We then determine the risk-reduction potential of these times, by defining their spatial relationship with exposure, event-dependent vulnerability, and an alert accuracy proxy, using well-established risk-prediction tools from earthquake engineering” has been changed to: “We then determine the potential usefulness of these times for EEW purposes, by defining their spatial relationship with exposure, seismic hazard, and an alert accuracy proxy, using well-established earthquake-engineering tools for measuring earthquake impacts.”*
- *The last line of the abstract has been changed from “The results are quantitative EEW feasibility maps, which demonstrate that EEW could be an effective risk-mitigation tool for a significant portion of Europe” to “The mapped feasibility results demonstrate that, under certain conditions, EEW could be effective for some parts of Europe.”*
- *The last sentence of the penultimate paragraph in the Introduction has been changed from: “We also explicitly quantify the risk-mitigation potential of these times for specific magnitude events, by establishing their spatial relationship with values of proxy measures for seismic vulnerability, exposure, and alert accuracy” to: “We also explicitly quantify the potential effectiveness of these times in the context of EEW, by establishing their spatial relationship with values of proxy measures for earthquake impact and alert accuracy.”*
- *We have changed the title of the “Risk Quantification” section to “Quantifying the Effectiveness of Computed Lead Times”*
- *We have modified the first line of the “Quantifying the Effectiveness of Computed Lead Times” section from: “We examine the risk-mitigation potential of the calculated lead times, by defining their spatial relationship with ambient (day/night) population distributions and the average seismic intensity...” to: “We examine the potential usefulness of the calculated lead times for EEW purposes, by defining their spatial relationship with ambient (day/night) population distributions and the average seismic intensity...”*
- *We have modified the description of population in the “Quantifying the Effectiveness of Computed Lead Times” section from: “Population is an important consideration in seismic risk assessment³⁶ that often acts as a proxy for the exposure (i.e., the value at risk) of the built environment/assets in earthquake engineering and risk modelling applications³⁷” to: “Population often acts as a proxy for the exposure (i.e., the value at risk) of the built environment/assets in earthquake engineering and risk modelling applications³⁷”*

- We have modified the description of seismic intensity in the “Quantifying the Effectiveness of Computed Lead Times” section from: “Seismic intensity, which describes the effect of earthquake ground shaking on the built environment and communities, is a well-established proxy for both damage³⁸ and vulnerability³⁹” to: “Seismic intensity describes the effect of earthquake ground shaking on the built environment and communities^{34,35}.”
- We have modified the first sentence of the fifth paragraph in the Discussion from: “The findings of this study suggest that an expansion/enhancement of Istanbul-focused EEW efforts could be significantly beneficial for mitigating seismic risk in many regions throughout Turkey” to: “In particular, the feasibility indices suggest that an expansion of EEW efforts in Turkey beyond Istanbul (by upgrading the hardware and software of existing strong-motion/broadband stations and networks for real-time data processing and telemetry capabilities) could be appropriate.”
- We have modified the third sentence of the fifth paragraph in the Discussion from: “In summary, we ultimately conclude that this work provides strong evidence to suggest that EEW could be an effective tool for supporting earthquake-related disaster risk reduction across a significant proportion of Europe” to: “We ultimately conclude that this work provides evidence to suggest that some parts of Europe could benefit from EEW as a helpful supplemental tool for supporting earthquake-related disaster risk reduction [3], but the extent of its effectiveness would be highly sensitive to the size of targeted events and the threshold at which an alert is triggered.”
- We have changed the title of the “Risk Modelling” subsection in Methods to “Lead-Time Effectiveness Modelling”

References:

- [1] Silva, V., Crowley, H., Varum, H., & Pinho, R. (2015). Seismic risk assessment for mainland Portugal. *Bulletin of Earthquake Engineering*, 13(2), 429-457.
- [2] Jaiswal, K., Wald, D., & Porter, K. (2010). A global building inventory for earthquake loss estimation and risk management. *Earthquake Spectra*, 26(3), 731-748.
- [3] Wald, D. J. (2020). Practical limitations of earthquake early warning. *Earthquake Spectra*, 36(3), 1412-1447.

- b. Since this paper is primarily about alert time (not warning times; point above). Quantification of actual risk reduction was not adequately presented, “risk” probably should not be in the Title.

To address this comment, we have changed the title of the study from: “Earthquake early warning could mitigate seismic risk across Europe” to “Could earthquake early warning be effective across Europe?”

- c. Discussion. “We further contextualised the significance of the lead times by combining them with spatial distributions of two risk proxies often used in earthquake engineering (i.e., population and seismic intensity).” Again, one cannot relate population and intensity to risk without knowing/specifying/estimating vulnerability and inventory.

We agree with the reviewer and have changed the highlighted sentence to: “We further contextualised the significance of the lead times by combining them with spatial distributions of two proxies often used to measure the effects of an earthquake in earthquake engineering (i.e., population and seismic intensity).”

- d. “... during this [warning] time, actions can be taken to significantly decrease detrimental impacts”. This is stated as if fact, yet no examples have been provided of evidence of decreased detrimental impacts. Aren’t there examples given all the existing EEW

systems mentioned? It is not defensible to make a general statement without citation to such examples? Simply citing others who make the same assertion in the EEW literature is not satisfactory. Perhaps the “*significantly decrease detrimental impacts*” part can be qualified or specified to make this statement scientifically accurate (*potentially, possibly, might*).

Thanks for this comment. To address this comment, we have modified the phrase “to significantly decrease detrimental impacts” to “that might decrease detrimental impacts”.

No examples of decreased detrimental impacts are provided in the abstract, given the restricted word limit (150 words), and no references are provided in the abstract, in accordance with the instructions of the journal.

We do provide some examples of the detrimental impacts that might be decreased with EEW in the first paragraph of the introduction, as follows: (1) Performing “drop, cover and hold on” (DCHO) [1] or moving to a safer location (either within a building or outside), to avoid injuries; (2) slowing down high-speed trains, to reduce accidents [2]; (3) shutting off gas pipelines, to prevent fires [3]; and (4) switching signals to stop vehicles from entering vulnerable infrastructure components (such as bridges), to avoid fatalities [4]. We also note that: “This list accounts for merely a small number of the vast array of critical applications that can benefit from EEW[5], and interested readers are referred to Wald [6] for a more thorough and critical discussion of this issue”.

References:

- [1] Porter, K. A. (2016). *How many injuries can be avoided through earthquake early warning and drop, cover, and hold on?*. Structural engineering and structural mechanics program, Boulder, CO.
- [2] Fabozzi, S., Bilotta, E., Picozzi, M., & Zollo, A. (2018). *Feasibility study of a loss-driven earthquake early warning and rapid response systems for tunnels of the Italian high-speed railway network*. *Soil Dynamics and Earthquake Engineering*, 112, 232-242.
- [3] Gasparini, P., Manfredi, G., & Zschau, J. (2011). *Earthquake early warning as a tool for improving society’s resilience and crisis response*. *Soil Dynamics and Earthquake Engineering*, 31(2), 267-270.
- [4] Le Guenan, T., Smai, F., Loschetter, A., Auclair, S., Monfort, D., Taillefer, N., & Douglas, J. (2016). *Accounting for end-user preferences in earthquake early warning systems*. *Bulletin of Earthquake Engineering*, 14(1), 297-319.
- [5] Velazquez, O., Pescaroli, G., Cremen, G., & Galasso, C. (2020). *A Review of the Technical and Socio-Organizational Components of Earthquake Early Warning Systems*. *Frontiers in Earth Science*, 8, 445.
- [6] Wald, D. J. (2020). *Practical limitations of earthquake early warning*. *Earthquake Spectra*, 36(3), 1412-1447.

- e. **There is almost no discussion about the connection of the hazard level (PGA and intensity) and what actions may be taken at each level that will reduce losses at those levels.**

To address this comment, we now mention in the discussion possible EEW actions that could be taken to reduce losses for the average seismic intensities experienced by the exposed population during events with a recurrence interval of 500 years, as follows:

“We found that almost all (i.e., approximately 98%) of the affected ambient population are exposed to average seismic intensities from large earthquakes at nearby sources that result in some falling objects, suggesting, for example, that EEW could help to protect against injuries through DCHO, evacuation or other means.”

- f. All the above lines of reasoning, the caption for Figure 7 must be revised. (“Examining the risk mitigation potential of calculated lead times. Average EMS-98 macroseismic intensities experienced by the affected ambient population during events with a recurrence interval of 500 years that resulted in at least 0.05 g PGA at the associated target site, categorised by the corresponding times of the maps presented in Figure 4. Note that seismic intensities V, VI, and VII denote “strong”, “slightly damaging”, and “damaging” events, respectively”). Again, this has no bearing on risk. The exposure of a person to a given shaking has no bearing on their risk: They could be outside, in a car, in a safe structure, and at shaking levels as low as 2 or 5%g they are not risk in nearly any situation (See below).

Thanks for this comment, which we agree with. To address it, we have removed the reference to risk in the caption of Figure 7. The caption now reads: “Examining the potential effectiveness of calculated lead times for EEW. Average EMS-98 macroseismic intensities experienced by the affected ambient population during events with a recurrence interval of 500 years that resulted in at least 0.05 g PGA at the associated target site, categorised by the corresponding times of the maps presented in Figure 4. Note that seismic intensities V, VI, and VII denote “strong”, “slightly damaging”, and “damaging” events, respectively.”

6. EEW Feasibility Modelling. The so-called “relative feasibility index measure” can take on almost any values depending on the weighting and the choices of population and intensity (PGA in this case) parameters chosen. I’m not sure how valuable this metric is either in an absolute or a relative sense. The authors note that “The indices of some target sites were significantly reduced (by up to almost 90%) in this case. However, this outcome is highly dependent on the assumed alert threshold (i.e., 0.05 g)²³. Using an alternative threshold of 0.02 g, for example, would have increased the accuracy of the worst-performing source-site combination by more than 20 times, due to less missed alerts.” The answer to the question of this study (EEW feasibility) is completely dependent on the choice of this weighting, and more specifically, on the threshold of PGA chosen. This means any outcome is possible and can be dialed-in based on the assumptions.

We agree that the relative feasibility index metric can change value, depending on the weighting applied to its different components; we believe that this is in fact an advantageous feature of the metric that makes it useful for decision-making purposes, as its value can change to reflect varying priorities of different stakeholders. It is not true to say that the metric can take on any value depending on the population and intensity parameters chosen – the index measures relative feasibility across different sites, which will not be significantly affected by the type of population/intensity parameter chosen (since the selected metric will be consistent across all evaluated sites).

We realise that simply multiplying the index by the proportion of correct alerts above a certain threshold leads to arbitrary outcomes and also distorts the ability of the metric to measure “relative” feasibility. We have therefore modified the “alert-outcome” version of the index to account for the relative performance of target sites in terms of alert accuracy. The modified formulation of this version of the index may be expressed as:

$$RF_{j,alert} = RF_j + F_{CA} (ca_j) \times w_{CA}$$

where $F_{CA} (ca_j)$ is the empirical cumulative distribution of the proportion of correct alerts at site j w_{CA} is its corresponding weight. It is important to note that the top three countries for EEW feasibility do not change depending on whether the triggering PGA is set to 0.02g, 0.05g, 0.1g, or 0.2g (at least in the case where all features of the equation are equally weighted).

7. Intro. “For each of these area sources, we calculate potential lead times (i.e., warning times that would be provided by EEW before shaking occurs) at target sites where the predicted median peak ground acceleration (PGA) associated with 500-year recurrence-interval events exceeds 0.05 g (see Figure 2b and Methods section), which is a commonly used threshold value for strong earthquake

shaking in several engineering applications, including seismic design aimed at life-safety performance^{28–30}.” You consider 5%g “strong shaking”, ShakeMap considers this “moderate” (Worden et al., 2020, figure below). First, if 5%g is strong, there are few adjectives left for 10, 20, 40 and 80%g? More to the point, selecting such a low threshold (5%g) for risk-reducing shaking makes EEW appear to be much more useful that it would be if you considered more reasonable (higher) shaking levels where risks are more obvious, but for which warning times are much shorter (e.g., Minson et al, 2019).

Thanks for this comment. We have addressed it as follows:

1. *We have modified our description of 0.05g from “strong earthquake shaking” to “moderate earthquake shaking”.*
2. *We now also examine the distributions of lead times for sources that exclusively produce median PGA predictions of at least 10%g at target sites (within the “Lead-Time Sensitivity Analyses” section of the revised manuscript). Figure 4 below shows the minimum, median, and maximum lead-time maps for this case. Our comments on these calculations can be found in the “Lead-Time Sensitivity Analyses” section of the revised manuscript as follows: “Less than 1% of these sites have positive minimum or median lead times (which are all smaller than 5 seconds, in both cases), and 17% have positive maximum lead times (<1% between 5 and 10 seconds, and the remainder less than 5 seconds).” The findings of these calculations contribute to the following conclusion made in the Discussion section: “... the viability of EEW is highly dependent on the magnitude of the ongoing event and the threshold PGA at which an EEW alert is issued at a target site.”*

Figure 4: (a) Minimum, (b) Median, and (c) Maximum lead-time maps for a 0.1g EEW alert threshold.

*Moreover, a shaking value of 0.02g (low end of intensity IV) is too low for any risk considerations, as is 0.05g (low end intensity V). See the legend below from ShakeMap (Worden et al., 2020). If one chose a higher threshold of shaking, where damage actually occurs, then the results of this study would be much less favorable in terms of average alert times. Note: Minson (2019) also studied thresholds of EEW and considered four PGA levels a user might want to receive warning: 2%g, 5%g, 10%g, and 20%g. These correspond roughly to intensity levels IV (light shaking) through VII (very strong shaking). Note further that if life safety is a consideration for EEW, then one must again double the PGA level to 40%g for intensity VIII, where, by definition structural collapse *begins* and fatalities are almost always associated with collapse of buildings. In Japan, JMA warns the population for JMA intensity 5L for its public warnings. Intensity 5L is *at least* IMMI VI, or closer to (IMMI VII) depending on the conversion scale used, so somewhere around 10%g to 20%g.*

Thanks for this comment. We realise that 0.05g may be too low a threshold for certain stakeholders and we have therefore addressed this comment by providing an additional analysis of lead times that focuses only on sites where

the PGA exceeds 0.1g for a given event (please see our previous response). Note that very high thresholds of PGA are not useful in the case of EEW, as EEW will not be able to mitigate the effects of building collapse.

Additional Comments.

1. Abstract: *“to an earthquake’s nucleation seconds to minutes before strong shaking occurs at target sites.”* “Minutes” (plural!) before is potentially applicable only to distant events; since you do not consider offshore events, the lead times you present can’t credibly include “minutes”, unless you are considering shaking levels well below societal relevance. This statement affords the critical reader to be immediately skeptical of what is to follow.

Thanks for pointing this out. We have removed the phrase “to minutes” throughout the manuscript, in line with the reviewer’s suggestion.

2. Introduction. Reference 2 (UNISDR. 2009) refers to early warning as in rapid alerting; this is not the same usage of the term *early warning* as used in *earthquake early warning*. In general, international multi-hazard agencies refer to quick reporting and impact assessment as “*early warning*”.

The complete definition of early warning from this reference (page 12) is: “The set of capacities needed to generate and disseminate timely and meaningful warning information to enable individuals, communities and organisations threatened by a hazard to prepare and to act appropriately and in sufficient time to reduce the possibility of harm or loss.” We believe that this definition is not just restricted to “quick reporting and impact assessment”, but also covers the concept of earthquake early warning.

During a conference on earthquake early warning systems at UNESCO in 2015, the Chief of the Regional Programme and Disaster Risk Reduction Section at UNISDR (Neil McFarlane) emphasised this definition in the context of earthquake early warning when he said: “There are three components to early warning. One is that you need to have access to good quality information that’s reliable and that people will trust. Secondly, it has to be timely. The third area is around the ‘last mile’, how do you get the information to the people on the ground quick enough.” A report containing this quote can be found here:

<https://www.undrr.org/news/new-earthquake-early-warning-drive-launched>

For these reasons, we have decided to keep the UNISDR reference in the manuscript.

3. Intro. *“The process of EEW typically involves up to five main steps: (1) Detecting an earthquake, (2) estimating its location; (3) estimating its magnitude; (4) estimating the ground-motion at target sites; and (5) using all of the information collected to decide whether (or not) to trigger an alarm. EEW systems may be broadly categorised as “regional”, “on-site” (or “hybrid”) depending on their approach to the first three steps mentioned above.”* Strictly speaking, on-site EEW systems do not actually need step 3: They typically predict the S wave amplitude based on P wave arrival (amplitude and period).

Thanks for this comment. “depending on their approach to the first three steps mentioned above” was a typo; the phrase should have been “depending on their approach to the first four steps mentioned above”. This correction has been made in the revised manuscript.

4. Intro. Ref #10, I don’t believe that Crowley et al. are specifically describing EEW as their strategy for risk reduction.

This is correct. However, the reference is used here to highlight the general importance of reducing seismic risk in the region; we do not imply that this risk should necessarily be reduced using EEW.

5. Intro. *“this is underlined by the fact that the average annual European GDP affected by earthquakes exceeds \$20 billion1”*. RE: Reference #11. Please explain how EEW is to reduce any financial (losses) risk as is implicit in your statement. It has not (yet at least) been shown to be true. Or, if so, provide specific examples.

Reference #11 is used here to highlight the general importance of reducing seismic risk in the region; we do not imply that this risk should necessarily be reduced using EEW.

6. Intro. *“This study therefore offers a unique trans-national perspective on the potential of EEW that is highly relevant for intergovernmental stakeholders, such as the International Search and Rescue Advisory Group (INSARAG) of the United Nations27.”* I’ve briefed INSARAG USAR teams numerous times. I find it questionable—or at least quite premature—to suggest that INSARAG is a EEW “stakeholder”. If I am wrong about this then please provide specifics.

We agree that it was premature of us to refer to INSARAG as an EEW stakeholder. To address this comment, we have refined the highlighted sentence to: “This study therefore offers a unique trans-national perspective on the potential of EEW that is relevant for intergovernmental bodies - such as the International Search and Rescue Advisory Group (INSARAG) of the United Nations27 – who may be interested in leveraging the technology.”

7. Using the notion of “*minimum lead time*” is a bit misleading. Any region affected by crustal earthquakes has blind zones for all existing EEW systems, particularly for the damaging moderatesized earthquake events.

To address this comment, we clarify that the “minimum lead time” for a given site implies the lowest lead time computed for that site in this study. This clarification is provided in the “Lead-Time Mapping for High Hazard Areas” section when the term is first introduced, as follows:

Figure 4 contains maps displaying the following three summary statistics for all affected target sites across the continent: (1) lowest computed lead time (i.e., “worst case scenario”), henceforth referred to as “minimum lead time”; (2) median computed lead time; and (3) largest computed lead time (i.e., “best case scenario”), henceforth referred to as “maximum lead time”.

8. Benchmarking the Calculations. *“The lead-time maps presented in Figure 4 are computed using a complex travel-time algorithm explicitly accounting for the physics of seismic wave propagation.* The terms “*complex*” and “*physics of seismic wave propagation*” seem a bit pretentious. Granted, you did not simply use a half-space approximation for travel times (as most do), but the calculations done are standard in other types of seismological analyses.

We appreciate this comment and have updated the highlighted sentence to: “The lead-time maps presented in Figure 4 are computed using a grid-based finite-difference travel-time algorithm.”

More generally, all other references to “sophisticated travel-time” and “complex travel-time” model/algorithm have been replaced with “grid-based finite-difference travel-time” model/algorithm, and all references to wave physics or wave propagation have been removed.

9. Benchmarking the Calculations. *“5% of minimum lead times, 16% of median lead times, and 64% of maximum lead times are over-estimated by at least one second when the simplified travel-time model is used, whereas 13% of minimum lead times, 8% of median lead times, and 4% of maximum lead times are under-estimated by at least the same amount.”* Given the generalized velocity structure employed, the use of “*complex, physics-based travel times*” is perhaps unnecessary. Other uncertainties in the calculation that were not considered are greater, for instance the evolution of

rupture for $M > 6.5$ events, requiring additional time to properly quantify; and the fact that the initial Pwaves and initial S-waves are not necessarily the pertinent arrivals. Likewise, the uncertainty in the delivery time is much greater, and that value has not even been considered.

We believe that a 1-D model is acceptable in this case, given that this type of model is routinely used in the NonLinLoc software for EEW purposes, i.e., the PRESTo platform (<http://www.prestoews.org/documentation.php>) mentioned by the reviewer. However, we appreciate that using a 1D model is nevertheless a limitation of our analysis, and we have added the following note in the Discussion section to highlight this simplifying assumption: “We used a 1-D velocity model in the travel-time algorithm, which does not capture lateral variations in the earth’s structure.”

Furthermore, we now account for uncertainty in the delivery time by incorporating a delay in our lead-time calculations that specifically accounts for the time required to compute different earthquake magnitudes. This delay time is assigned in line with [1]; it is assumed to equal 3 seconds for magnitudes less than 6.5, 4 seconds for magnitudes between 6.5 and 7, 12 seconds for magnitudes between 7 and 7.5, and 20 seconds for magnitudes greater than 7.5. The value of this delay used for a given area source is based on the magnitude of the event with a recurrence interval of 500 years (except when magnitude 5 events are examined, in which case δm is uniformly set to 3 seconds). We have added the following text on these delay times to “Lead-time Modelling” in Methods: δm represents the time required to compute the magnitude of the ongoing event, and is assumed to equal 3 seconds for $M_w < 6.5$, 4 seconds for $6.5 \leq M_w < 7$, 12 seconds for $7 \leq M_w < 7.5$, and 20 seconds for $M_w \geq 7.5$ [1]. The value of δm used for a given area source is based on the magnitude of the event with a recurrence interval of 500 years (except when M_w 5 events are examined, in which case δm is uniformly set to 3 seconds). Note that this delay is added to a 2-second data telemetry time to determine the total delay interval for a given lead-time calculation.

References:

[1] Trugman, D. T., Page, M. T., Minson, S. E., & Cochran, E. S. (2019). Peak ground displacement saturates exactly when expected: Implications for earthquake early warning. *Journal of Geophysical Research: Solid Earth*, 124(5), 4642-4653.

Figure 5 below shows the minimum, median, and maximum lead-time maps for the modified delay intervals considered in the updated manuscript. 3% of sites have positive minimum lead times, 18% of sites have positive median lead times, and 79% of sites have positive maximum lead time. Please see the “Lead-Time Mapping for High Hazard Areas” section of the manuscript for a more complete discussion of the results.

Figure 5: (a-c) Minimum, (d-f) Median, and (g-i) Maximum lead-time maps, assuming a delay interval of (a,d,g) 4 seconds; (b,e,h) 10 seconds; and (c,f,i) 20 seconds.

More generally, all references to “sophisticated travel-time” and “complex travel-time” model/algorithm have been replaced with “grid-based finite-difference travel-time” model/algorithm, and all references to wave physics or wave propagation have been removed.

We have decided to remove the benchmarking studies from the revised manuscript as, in line with the reviewer's concerns, we realise that these calculations somewhat detract focus from the more significant sources of uncertainty that exist in the calculations.

10. *"we introduce a novel feasibility metric that enables identification of priority regions". As an Editor myself, I find the use of terms like "novel" and "innovative" presumptuous on the part of authors. Judgement of novelty and innovative should be rendered by the reader.*

Thanks for this comment. We agree and have removed all mentions of "novel" from the manuscript. We have changed the only mention of "innovative" to "first", such that the last line of the Conclusions now reads as follows: "Despite these constraints, this study nevertheless represents a first attempt to comprehensively quantify EEW feasibility on a continental scale and to identify priority regions for more detailed EEW feasibility analyses/investment in EEW implementation."

11. Risk Quantification. *"98% of the total ambient population". Here and elsewhere in this paragraph, one should avoid beginning a sentence with a numerical value.*

To address this comment, we have replaced all numerical values at the start of sentences with equivalent words. For example, "98% of the total ambient population" has been modified to: "Ninety-eight percent of the total ambient population".

12. Risk Quantification. *How could higher intensities have longer lead times (higher percentage of pop. has > 10 sec of warning)? This conflicts with Minson's, Adrian-Meyer's, and all other studies, no?*

Our results simply imply that a higher proportion of the population live in areas that experience longer lead times for higher seismic intensities. This would only be counterintuitive (and therefore in conflict with the results of previous studies) if the population were uniformly distributed throughout the target sites examined (which is not the case).

13. Discussion. *"For example, they show that almost half (i.e., 44%) of the examined target sites benefit from warning times in a "best case scenario" that are long enough to accommodate major risk intervention actions, such as the shutting down of industrial equipment or the removal of vehicles from garages". First "could (or might) benefit" is more appropriate. Second, "removal of vehicles from garages" is not an action that I have ever come across as a potential benefit of EEW. Seems not so credible an action to me. Please provide a reference. Or, if you referring to opening garage doors at firehouses, that's more specific that the wording you chose?*

We have qualified the highlighted statement by adding the word "could" before "benefit", per the reviewer's suggestion.

The action "removal of vehicles from garages" was obtained from the following reference:

Goltz, J. D. (2002). Introducing earthquake early warning in California: A summary of social science and public policy issues. Caltech Seismological Laboratory, Disaster Assistance Division, A report to OES and the Operational Areas.

where it was reported as an example of an action that could be taken in 50 seconds, as part of a survey of potential Californian EEW users. This reference is reported in the caption of Table 1.

14. *It would be nice to have a sense for why Iceland warrants EEW according to this study? It would be useful to attempt to explain how Iceland came to the top of the list? Tectonics? What are the controlling factors that make other countries more or less suitable for EEW?*

This comment is no longer completely relevant as Iceland is not one of the most feasible countries for EEW according to the modified calculations of the revised manuscript. The controlling factors that make countries more or less suitable for EEW are examined by varying the weights applied to the different variables included in the relative feasibility index calculation (please see Figures 8b-d of the revised manuscript). For example, it is found that weighting lead time the most causes the largest feasibility increase at a site in Georgia and decrease at a site in Italy, weighting intensity the most causes the largest feasibility increase at a site in Croatia and decrease at a site in Georgia and weighting population the most causes the largest feasibility decrease at a site in Bosnia and Herzegovina and increase at a site in Greece.

15. Discussion. *“We did not explicitly consider the performance of existing EEW algorithms in the leadtime calculations. Instead, we simply assumed that four seconds was sufficient to capture both data telemetry delays and the length of time required by the algorithms to estimate relevant source parameters.”* This is a rather essential assumption that affects the results. This should be stated much earlier in the manuscript.

Thanks for pointing this out. Our lead-time calculations now explicitly incorporate separated (1) magnitude-dependent delay times reported in the literature for computing the magnitudes of ongoing events and (2) data telemetry time. (please see more details on these calculations in our response to Additional Comment #9).

16. Competing interests. *“The authors declare no competing interests”.* I’m not sure I completely agree with this declaration. The authors state note:

“A new three-year Horizon 2020 European research project called TURNkey (Towards more Earthquake-resilient Urban Societies through a Multi-sensor-based Information System enabling Earthquake Forecasting, Early Warning and Rapid Response actions; see Acknowledgements section) aims to address this issue by developing a holistic earthquake information system that incorporates state-of-the-art seismic risk mitigation tools for both operational earthquake forecasting and EEW in real- and near-real time, with selected testbeds in Italy and Greece (as well as Iceland) to be the focus of more detailed analyses for EEW and the target of end-userorientated applications of the system.”

The Acknowledgements note that this study was *funded* by Horizon 2020. The statement quoted above is quite clearly advocacy for Horizon2020/Turnkey. Is that *not* a competing interest?

*This is not a competing interest. The point of this study was to impartially assess the potential usefulness of using EEW across Europe, and **the source of our funding does not affect the integrity of this research in any way; it is important to underline that we are under no obligation from our funding source to come to any particular conclusion on this work.** We are upstanding researchers, whose goal is always to advance the state of knowledge in an honest, objective and scientifically rigorous manner.*

The TURNkey project is committed to objectively investigating the feasibility of EEW, rather than to promoting or “selling” its use. In addition, it is important to note that TURNkey focuses on an entire suite of real- and near-real-time earthquake loss-mitigation aids (i.e., operational earthquake forecasting, EEW and rapid response actions) for Europe, each of which is being investigated for use with an equal amount of effort and focus.

However, we agree with the reviewer that the highlighted sentence may have provided a misleading impression to readers that a conflict of interest does exist. We have therefore removed this sentence from the manuscript.

Reviewer #3 (Dr. Yannick Behr):

In this study, the authors present a probabilistic evaluation of the usefulness of earthquake early warning in Europe, focusing on the Euro-Mediterranean region with some mentions of Iceland. Starting from the 2013 European Seismic Hazard model, they develop a novel measure to quantify the usefulness of EEW

for a particular target site based on potential lead times, population density, and the ground acceleration for which a warning would be possible.

In the following I list major points that I think need to be addressed before publication. I have added further, minor comments to the manuscript itself.

We appreciate the reviewer's comments and have made significant changes to the manuscript in response. Please note that we have included responses to the minor comments at the end of this document, for convenience.

1. In my opinion the authors overestimate the available lead time. One aspect of this is the inclusion of temporary seismic stations in the network analysis and the assumption that telemetry delays do not exceed 1-2 s. These are quite idealized conditions [1] and I think this should be made clearer in the main text. It is briefly mentioned in the discussion but I think it deserves more attention. Getting a seismic network EEW ready is a long and costly undertaking.

Thanks for this comment. To address it, we have removed temporary seismic stations from our analysis. The only seismic stations considered in the calculations of the revised manuscript are permanent broadband and strong-motion stations from the IRIS database.

While the stations here are not transmitting data in real time, we are assuming that they could be upgraded for use in an EEW system (as mentioned in the part of the Discussion section highlighted by the reviewer). We therefore assume data telemetry delay times that correspond with real-time communication capabilities (as opposed to current system configurations). These delays are now explicitly incorporated in the lead-time calculation (in addition to magnitude-dependent delays associated with computing the magnitude of ongoing events -see response to Comment#2), and are assumed to comprise of 1 second for data transmission and 1 second for issuing the warning message, which is directly in line with previous EEW studies [3-5]. We comment on the ideal nature of this delay in two places:

3. "Lead-Time Mapping for High Hazard Areas" section now contains the following line: "*These calculations incorporate a magnitude-dependent delay interval that captures the time taken to compute characteristics of the ongoing event and to complete a state-of-the-art real-time data telemetry process (see Methods section).*"
4. "Lead-time Modelling" in Methods "*[The data telemetry time] captures data telemetry delays, which are idealistically assumed to comprise 1 second for data transmission and 1 second for issuing the warning message [3-5].*"

References:

- [1] Auclair, S., Goula, X., Jara, J. A., & Colom, Y. (2015). Feasibility and interest in earthquake early warning systems for areas of moderate seismicity: Case study for the Pyrenees. *Pure and Applied Geophysics*, 172(9), 2449-2465.
- [2] Thelen, W. A., Hotovec-Ellis, A. J., & Bodin, P. (2016). Feasibility study of earthquake early warning (EEW) in Hawaii (No. 2016-1172). US Geological Survey.
- [3] Allen, R. M. (2007). The ElarmS earthquake early warning methodology and application across California. In *Earthquake early warning systems* (pp. 21-43). Springer, Berlin, Heidelberg.
- [4] Behr, Y., Clinton, J., Kästli, P., Cauzzi, C., Racine, R., & Meier, M. A. (2015). Anatomy of an earthquake early warning (EEW) alert: Predicting time delays for an end-to-end EEW system. *Seismological Research Letters*, 86(3), 830-840.
- [5] Zuccolo, E., Gibbs, T., Lai, C. G., Latchman, J. L., Salazar, W., Di Sarno, L., ... & Workman, A. (2016). Earthquake early warning scenarios at critical facilities in the Eastern Caribbean. *Bulletin of Earthquake Engineering*, 14(9), 2579-2605.

- Further, in their lead time calculation, the authors assume that the final magnitude can be estimated from the first 1-2 s of the P-wave. With an additional 1-2 s for data telemetry delays and processing time, they assume that the final magnitude is known after a 4 s interval once a third station has detected the P-wave. Several studies have shown, however, that magnitude estimates from EEW algorithms using only the first few seconds of the P-wave recording tend to saturate at magnitudes greater than Mw 6.5 [2,3]. The likely reason is that rupture onset between an earthquake with Mw \leq 6.5 and Mw $>$ 6.5 is indistinguishable during the first 4 seconds of the rupture [4]. For an Mw 8 it may even take up to 20 s after the rupture initiation before an EEW system could estimate the magnitude accurately [5]. As a consequence, targets further away from the epicenter do not necessarily have longer lead times [5]. I think this is an important aspect that the authors could include in their lead time calculations without too much additional work.

To address this comment, we now incorporate a delay in our lead-time calculations that specifically accounts for the time required to compute different earthquake magnitudes. This delay time is assigned in line with [1]; it is assumed to equal 3 seconds for magnitudes less than 6.5, 4 seconds for magnitudes between 6.5 and 7, 12 seconds for magnitudes between 7 and 7.5, and 20 seconds for magnitudes greater than 7.5. The value of this delay used for a given area source is based on the magnitude of the event with a recurrence interval of 500 years (except when magnitude 5 events are examined, in which case δm is uniformly set to 3 seconds). We have added the following text on these delay times to “Lead-time Modelling” in Methods: “ δm represents the time required to compute the magnitude of the ongoing event, and is assumed to equal 3 seconds for Mw $<$ 6.5, 4 seconds for $6.5 \leq$ Mw $<$ 7, 12 seconds for $7 \leq$ Mw $<$ 7.5, and 20 seconds for Mw \geq 7.5 [1]. The value of δm used for a given area source is based on the magnitude of the event with a recurrence interval of 500 years (except when Mw 5 events are examined, in which case δm is uniformly set to 3 seconds). Note that this delay is added to the 2-second data telemetry time (see our response to Comment #1 for more details) to determine the total delay interval for a given lead-time calculation.

References:

[1] Trugman, D. T., Page, M. T., Minson, S. E., & Cochran, E. S. (2019). Peak ground displacement saturates exactly when expected: Implications for earthquake early warning. *Journal of Geophysical*

Figure 6 below shows the minimum, median, and maximum lead-time maps for the modified delay intervals considered in the updated manuscript. 3% of sites have positive minimum lead times, 18% of sites have positive median lead times, and 79% of sites have positive maximum lead time. Please see the “Lead-Time Mapping for High Hazard Areas” section of the manuscript for a more complete discussion of the results.

Figure 6: (a) Minimum, (b) Median, and (c) Maximum lead-time maps resulting from the updated lead-time calculations.

- I am not completely convinced by the merit of the Relative Feasibility Index with the focus here on “Relative”. If I understand correctly, in the hypothetical case that the maximum median lead time

across all target sites was 2 s, a site with a median lead time of 1.9 s would have a high score even though the lead time would be too short for most mitigation actions. I think a function expressing the actual usefulness of an EEW alert would be more informative than the empirical cumulative distribution function.

Yes, it is correct to say that a site with a median lead time of 1.9 seconds would have a high score in the case you described. However, we can easily account for the “non-feasibility” of sites with lead times less than two seconds by removing these sites from the feasibility calculations. In fact, we do something similar in our calculations by only considering the relative feasibility of sites that have a positive median lead time. The advantage of a relative approach is that it enables direct comparisons between the feasibility of different sites, without requiring a subjective interpretation of the “usefulness” of an EEW alert; we have therefore decided to keep the index in the original format. However, we realise that simply multiplying this metric by the proportion of correct alerts above a certain threshold leads to arbitrary outcomes and also distorts the ability of the metric to measure “relative” feasibility. We have therefore modified the “alert-outcome” version of the index to account for the relative performance of target sites in terms of alert accuracy. The modified formulation of this version of the index may be expressed as:

$$RF_{j,alert} = RF_j + F_{CA}(ca_j) \times w_{CA}$$

where $F_{CA}(ca_j)$ is the empirical cumulative distribution of the proportion of correct alerts at site j and w_{CA} is its corresponding weight.

4. I think the authors spend too much time in the discussion on reiterating their results and I missed a discussion on what would be necessary to improve EEW capability in the Euro-Mediterranean region.

Thanks for this comment. We have extensively modified the discussion section in response to this comment. These modifications may be summarised as follows:

- 1. We have significantly shortened the text that reiterates the results. For example, we have removed descriptions like: “Probabilistic distributions of EEW lead times were determined for target sites in these areas, by randomly varying the location of seismic sources in line with regional seismicity.”*
- 2. We now explicitly discuss components that dictate the extent of EEW capability in the region, such as the PGA threshold at which alerts are triggered and the size of targeted seismic events.*
- 3. We include more discussion on what is necessary to improve EEW capability. We mention that Turkey could benefit from an expansion of its Istanbul system “by upgrading the hardware and software of existing strong-motion/broadband stations and networks to real-time data processing and telemetry capabilities”. As in the original version of the manuscript, we explicitly state that EEW capability could be improved in regions with the lowest lead times “through increased seismic station density”. Note that consideration of the costs required to build and maintain EEW systems is outside the scope of this paper, as mentioned in the last paragraph of the discussion section.*

Overall, the manuscript is well written and contains appropriate and balanced references to previous work. I believe it would be possible to reproduce results given the information in the manuscript. In my opinion, it is an important contribution to an ongoing discussion about the usefulness of earthquake early warning and contains some novel approaches to help answer this question.

We appreciate the reviewer’s positive assessment of our work.

Additional Comments

1. I think 10s of seconds would be more accurate. I doubt that EEW can provide minutes of early warning before strong shaking.

To address this comment, we have removed the phrase “to minutes” throughout the manuscript.

2. The study mostly focuses on this region so it may be more accurate to reflect that in the title and the first sentence of the abstract as well as in the discussion.

According to https://en.wikipedia.org/wiki/Euro-Mediterranean_region, the Euro-Mediterranean region “encompasses all the European Countries and the countries on the Mediterranean rim”. It appears from other comments that the author may instead interpret this region as that which exclusively encompasses Mediterranean countries.

We stick to use of the term “Europe” in the title and the first sentence of the abstract, since the vast majority of examined countries are European.

3. This graph is a bit difficult to read on a A4 hardcopy. Maybe replace with a density plot.

Thanks for this suggestion. We produced a station density plot (provided below in Figure 7), in line with this suggestion, but we ultimately feel that this is more difficult to interpret than the original station plot.

Figure 7: Density plot of seismic stations considered in this study.

Therefore, we have opted to maintain the original station plot (provided in Figure 8 below), except that we now distinguish between considered strong-motion and broadband stations.

Figure 8: Plot of seismic stations considered in this study (i.e., Figure 1a of the revised manuscript).

4. I think it would be better to focus these and the following maps on the Euro-Mediterranean region as this is also the focus of the current study.

Thanks for this comment. These maps are already focused on the Euro-Mediterranean region, although we believe the reviewer may be referring exclusively to the Mediterranean region (please see our response to Additional Comment #2). Since our study also encompasses Iceland, we believe it would be misleading to focus our maps exclusively on the Mediterranean region, and have maintained the same map extents in the revised version of the paper.

5. This section could be shortened or be incorporated in the previous section.

This is a great suggestion. In fact, we have completely removed this section in response to another reviewer's comment.

6. I don't think this figure is needed. A quantitative comparison rather than a visual comparison with Figure 4 would be better.

We have removed the two benchmarking figures (and indeed the complete set of benchmarking calculations) from the revised version of the manuscript, in response to another reviewer's comment. However, we provide a quantitative comparison of the lead-times in Table 1 below (note that delay times related to magnitude computation and data telemetry are ignored in this case). We can conclude from this comparison that there is an acceptable level of alignment between the results obtained for both resolutions.

Table 1: Verifying the adequacy of the resolution chosen for the lead-time maps by comparing the lead-times obtained for Italy with those that result from using a finer seismic source/target site grid spacing.

Lead Time Map and Time Range	Proportion of high-resolution	Proportion of coarse-resolution
-------------------------------	---------------------------------

	sites within time range	sites* within time range
Minimum lead times (0-5 seconds)	0.17	0.19
Minimum lead times (5-10 seconds)	0.06	0.05
Minimum lead times (>10 seconds)	0.01	0.00
Median lead times (0-5 seconds)	0.58	0.61
Median lead times (5-10 seconds)	0.35	0.33
Median lead times (>10 seconds)	0.05	0.03
Maximum lead times (0-5 seconds)	0.03	0.05
Maximum lead times (5-10 seconds)	0.40	0.47
Maximum lead times (>10 seconds)	0.57	0.48

**The unique set of examined sites that are located closest to the sites of the high-resolution Italian map.*

7. I don't understand why Iceland is discussed here but Portugal isn't.

This comment is no longer relevant, as the results of this section have changed according to our modified calculations. We discussed Iceland here, because it contained sites with higher feasibility than Greece, and we wanted to include Greece to emphasise that the results did not significantly change when alert accuracy was accounted for.

8. The first part of the discussion is mainly a repeat of the results sections. I don't think that's necessary.

We have addressed this comment by shortening/removing text in the discussion section that reiterates the results. For example, we have removed descriptions like: "Probabilistic distributions of EEW lead times were determined for target sites in these areas, by randomly varying the location of seismic sources in line with regional seismicity."

9. Black triangles are very hard to see and can be easily interpreted as dots with Relative Feasibility Index = 1. Probably better to zoom into the Euro-Mediterranean region and use a different color for the triangles.

Thanks for this suggestion, which we have implemented by replacing the black triangles with bright green triangles. See Figure 9 below for an example of the modified Relative Feasibility Index plots. Note that we have kept the European-wide extents of these plots, so that the results for Iceland are visible.

Figure 9: Sample Relative Feasibility Index plot included in the modified version of the manuscript

10. “and and” typo highlighted in the text.

Many thanks for spotting this typo, which has been amended in the revised version of the manuscript.

REVIEWER COMMENTS

Reviewer #1 (Remarks to the Author):

Dear Editor, I appreciated the improvements done to account for my comments and for the comments of the other reviewers. However, I recognize the many of raised questions have not been tackled with supporting analysis, and are only commented in the discussion section, as actual limitations of the manuscript.

I still think that some problems and limitations raised in the previous review have not been addressed and significantly affect the results.

The station distribution used here does not reflect the real landscape of strong motion stations in Europe. The IRIS catalog deserves diverse objectives as compared to the evaluation of the ground shaking; this is even more true for Europe. Several institutions are not committed to share station information with IRIS, although their stations can be used and are valuable for Early Warning. Beyond the case of Spain with almost no stations in the IRIS catalog, other very relevant stations are missing in this catalog. This is the case of the Italian National Strong motion network IT – operated by the Italian Civil protection – and the Turkish National Strong Motion Network TK – operated by AFAD (Compare stations at <https://deprem.afad.gov.tr/istasyonlar?lang=en> with stations reported in the paper). Turkey and Italy are two main targets of the study. At this stage, I would have preferred to have an average estimation of the inter-station distance per small areas and use this information for the analysis, independently of the effective location of the stations (also because real lead-time of stations is not evaluated).

Furthermore, the authors still use broadband sensors, probably since the IRIS database is mainly populated by those sensors. However, saturation associated to the sensor mechanics, sensitivity and datalogger clipping could be very relevant for large magnitude events. This limitation can affect the use of those sensors in the near-source domain. This issue could be addressed in the manuscript, first removing these sensors, then considering a reliable selection of sensors based on magnitude/distance criteria.

The authors now consider the saturation of the Early warning indicators with magnitude and they expand the time window, to account for correct magnitude estimation of large events. However, it seems to me that they only assume a variable delay as a function of magnitude, which is not correct for regional Early warning systems. Within this strategy, S wave arrives at close stations well before having 12s or 20s of P wave time window, and those stations need to be not considered for magnitude estimations. Thus, following this simplistic approach, we should only use stations located enough far to catch 12s or 20s of P wave, avoiding the pollution of S waves.

Finally, I apologize to have provided the impression that I do not appreciate a 1D model. I do appreciate 1D models even for travel-time computation. I just point out that a uniform 1D model – CRUST 1.0 – based on long period arrival times – is not considered as a reliable model for local measurements of travel-times. Regional Early Warning techniques use 1D models, but calibrated for the target area. Thus, a collection of diverse 1D models, for the different tectonic regions, will provide a significantly more accurate velocity model, that would be an effective added value for this study.

Reviewer #2 (Remarks to the Author):

I give the authors an enormous amount of credit for their successful and comprehensive revision (and suitable rebuttals in a few other cases). I reviewed the reconciliation to the other two reviewers' comments and those too were addressed comprehensively by the authors.

In this case, the peer-review process worked as hoped. The revision is reframed (right down to the title) to the extent that it is a much more balanced and valuable paper than the original. It is ready for publication and will make a valuable contribution to this topic, likely leading the EEW discussion in Europe.

There are two minor quibbles that I mention below, that do not require attention at this time, but the authors might mull over:

#5 "the average annual European GDP affected by earthquakes exceeds \$20 billion". I still think that juxtaposing total earthquake losses with EEW implicitly makes the case that those losses could be reduced by successful EEW. I would provide a caveat there.

Item #13, concerning "removal vehicles from garages". Since the study that recommended that gave interviewees 50-sec warning, and asked what they could do with 50 sec, that response came up. It seems that this is no longer a practical consideration given what we've learned in this study so you might consider eliminating that as a plausible mitigation strategy. Going into a garage for this express purpose seems like a bad idea.

Reviewer #3 (Remarks to the Author):

The manuscript with the title "Could earthquake early warning be effective across Europe" has much improved by the significant amount of work the authors put in to address the reviewers comments. Reported lead times seem more realistic and, by adding additional magnitude dependent delays, are more in line with recent EEW research. Also, many assumptions underlying the lead time calculations have been made more explicit. Overall the text reads well and has become more concise. I particularly welcome the fact that the authors are planning to publish the source code of their analysis.

The updated version has, however, increased my doubts about the usefulness of a relative feasibility index as presented by the authors. Although median lead times have decreased significantly in many regions, the feasibility index for regions with positive median lead times appears unchanged or even improved. I just do not understand how such an index could be useful to decision makers who have to decide whether or not to invest in EEW. To me it masks the fact that median lead times are, in the majority of the cases, between 0-5 s. In my opinion, deciding on an, albeit subjective, lead time that would be 'useful' and then measuring the feasibility based on that would provide a lot more value. If the authors decide to keep using the relative index I think they should make it crystal clear how to interpret the index. The easiest way to do that would be by adding a few examples for various indices (e.g. 1, 0.8, 0.2). The authors added an additional, useful, explanation to the Methods section, but I think a few examples with actual values in the main text are needed too.

My other main point may seem like a technicality but I think it is quite important: with the updated lead times there is even more reason to focus the maps on the regions that are actually discussed in the manuscript. For example, Portugal shows a small area with positive median lead times but this is never mentioned in the text. In the updated analysis, Iceland doesn't have positive lead times anymore. I think it suffices to mention that the analysis was done for the whole Euro-Mediterranean region but that only countries will be discussed for which EEW seems feasible. There is a lot of wasted space in these maps which makes it also quite difficult to see details in the parts of the maps that matter.

The main message I take away from this study is that EEW with regional algorithms only has limited value for many places in Europe and that future work should maybe focus on trying to find different algorithms rather than focus on this one particular type. I think this is an important finding and contribution to an ongoing discussion about the feasibility and usefulness of EEW. I enjoyed being involved in this study as a reviewer and the thought-provoking comments by the authors and the other reviewers.

Yannik Behr

Response to comments for “Could earthquake early warning be effective across Europe?”, by Gemma Cremen, Carmine Galasso, and Elisa Zuccolo

NCOMMS-20-42346

We thank the reviewers for their thoughtful comments, which have further improved the quality of the revised manuscript. The reviewers’ comments have been numbered and listed below (in red), followed by our responses in italics.

Reviewer #1:

I still think that some problems and limitations raised in the previous review have not been addressed and significantly affect the results.

We again acknowledge the valuable comments of the Reviewer in the first round of review that ultimately led to a higher quality manuscript. We thank the Reviewer for their additional comments, which are addressed in detail below.

1. The station distribution used here does not reflect the real landscape of strong motion stations in Europe. The IRIS catalog deserves diverse objectives as compared to the evaluation of the ground shaking; this is even more true for Europe. Several institutions are not committed to share station information with IRIS, although their stations can be used and are valuable for Early Warning. Beyond the case of Spain with almost no stations in the IRIS catalog, other very relevant stations are missing in this catalog. This is the case of the Italian National Strong motion network IT – operated by the Italian Civil protection – and the Turkish National Strong Motion Network TK – operated by AFAD (Compare stations at <https://deprem.afad.gov.tr/istasyonlar?lang=en> with stations reported in the paper). Turkey and Italy are two main targets of the study.

We agree that an exhaustive search of local seismic station databases is warranted (and critical) for detailed, regional lead-time studies on a small scale. However, our study has a much broader scope, encompassing several entire countries on a continent-wide scale. We believe it is important to use consistent and open high-level data sources that appropriately reflect this, so that the study is fully replicable. This is why all of our data (including seismic hazard and population information) is sourced from open databases covering the continent or global scale. Leveraging local databases, in this case, may create an unfair bias against countries/regions that do not provide/store this level of information and raises the possibility of introducing discrepancies in the accuracy of data used. Therefore, we are not in favour of using additional seismic station information.

Furthermore, we note that other large-scale studies published in Nature Communications and alternative reputable journals have also exclusively relied on high-level (e.g., global-scale) databases[e.g., 1-4].

*To fully reflect the Reviewer’s concern, we have added a further note on the type of data used in this work to the last paragraph of the “Discussion” section, such that the first four sentences of this paragraph now read: “It is important to note that there are some limitations/simplifying assumptions associated with this work that warrant comment. Firstly, we leveraged an international database to obtain details on seismic stations across the continent (see **Methods** section). This approach may not have completely captured all stations across Europe, and our calculations may have underestimated actual lead times in some cases; for the actual design/implementation of EEW systems in any region, an exhaustive search of local databases would be critical to produce detailed and accurate lead-time estimates. However, exploiting local seismic station databases in this study could have created an unfair bias against countries/regions that do not provide/store this type of information and may have introduced discrepancies in the quality of information used. In fact, all of the data employed in this work (including those related to seismic hazard and population) are from consistent and open*

high-level sources to reflect the broad geographical extent of the study and ensure the results are fully replicable.”

References:

- [1] Koks, E. E., Rozenberg, J., Zorn, C., Tariverdi, M., Vousedoukas, M., Fraser, S. A., ... & Hallegatte, S. (2019). A global multi-hazard risk analysis of road and railway infrastructure assets. *Nature Communications*, 10(1), 1-11.
- [2] Koks, E. E., & Haer, T. (2020). A high-resolution wind damage model for Europe. *Scientific reports*, 10(1), 1-11.
- [3] van Ginkel, K. C., Dottori, F., Alfieri, L., Feyen, L., & Koks, E. E. (2021). Flood risk assessment of the European road network. *Natural Hazards and Earth System Sciences*, 21(3), 1011-1027.
- [4] Silva, V., Crowley, H., & Bazzurro, P. (2016). Exploring risk-targeted hazard maps for Europe. *Earthquake Spectra*, 32(2), 1165-1186.

At this stage, I would have preferred to have an average estimation of the inter-station distance per small areas and use this information for the analysis, independently of the effective location of the stations (also because real lead-time of stations is not evaluated).

We thank the Reviewer for this comment. We have taken a first attempt at addressing it, by interpolating interstation distances across the continent (using the IRIS stations of the current manuscript version; see Figure 1) and substituting these values as a proxy for lead times in the feasibility index (where smaller interstation distances indicate higher feasibility, in line with the findings of [1]). (Note that the interstation distance of a given seismic station is taken as the average distance to the closest three stations, per the original calculations of the paper). The results of the preliminary calculations are provided in Figure 2, which are limited to regions affected by interstation distances less than 30 km. The conclusions remain the same, i.e., Italy, Turkey, and Greece are still the countries with the highest feasibility. (Note that this finding was largely expected, since this alternative approach is still calibrated on the IRIS database).

However, formally implementing interstation distance instead of lead times in the calculations would lead to substantial changes in the manuscript's content (directly contradicting some of the comments of Reviewers #2 and #3); for example, the sections on “Lead-Time Mapping for High Hazard Areas” and “Lead-Time Sensitivity Analyses” (which collectively represent a significant portion of the existing manuscript) would need to be completely changed or removed. In any case, the interstation distance metric does not account for differences in velocity between areas, the exact locations of stations, or differences in computation time due to different magnitude events, so it is less informative than the lead time calculations provided. Therefore, we would prefer not to substitute lead times with interstation distance in our analyses.

To reflect the Reviewer's comment, we have added a note on the use of interstation distances as a proxy for lead time in the last paragraph of the “Discussion” section, as follows: “In any case, preliminary investigations indicate that the conclusions of the feasibility mapping do not strongly depend on the accuracy of the lead-time calculations; using interpolated values of interstation distance (see **Methods** section for details on this metric) as a proxy for lead times in the feasibility index (where smaller distances indicate higher feasibility, in line with the findings of Kuyuk and Allen (36)) still produces the largest index values in Turkey, Italy and Greece.”

References:

- [1] Kuyuk, H. S., & Allen, R. M. (2013). Optimal seismic network density for earthquake early warning: A case study from California. *Seismological Research Letters*, 84(6), 946-954.

Figure 1: Interpolated interstation distances at examined area sources across the continent (note that large interstation distances in regions with longitudes less than 18 are not shown, to enhance readability of the map in target areas of the study).

Figure 2: Equally weighted Relative Feasibility Index mapping, substituting median interstation distance (across all sources affecting a given target site) for median lead time. Note that the feasibility study is limited to sites where the interstation distance is less than or equal to 30 km.

2. Furthermore, the authors still use broadband sensors, probably since the IRIS database is mainly populated by those sensors. However, saturation associated to the sensor mechanics, sensitivity and datalogger clipping could be very relevant for large magnitude events. This limitation can affect the use of those sensors in the near-source domain. This issue could be addressed in the manuscript, first removing these sensors, then considering a reliable selection of sensors based on magnitude/distance criteria.

We thank the Reviewer for this comment. However, we have opted to keep the broadband sensors in our analyses for reasons outlined in the previous review, i.e.,:

- Broadband stations are used in operational EEW systems (such as ShakeAlert in the US) [1];
- EEW algorithms have been specifically designed for broadband stations [2].

Nevertheless, we clearly state our working assumption is that all considered stations will be upgraded for actual use in earthquake early warning if necessary (see Discussion section): “it is assumed that the considered seismic stations are (or could be) capable of being used for early warning purposes (i.e., they have or could have adequate data acquisition/transmission systems, real-time communication capability, robust dissemination methods, power supply systems, etc.)”, which is in line with previous earthquake early warning feasibility studies [e.g., 3].

It is also important to note that investigating the performance of EEW algorithms (including the criteria used for including/excluding data from broadband sensors) is not within the scope of this study – see our response to the next comment for more details.

References:

[1] Chung, A. I., Meier, M. A., Andrews, J., Böse, M., Crowell, B. W., McGuire, J. J., & Smith, D. E. (2020). ShakeAlert earthquake early warning system performance during the 2019 Ridgecrest earthquake sequence. *Bulletin of the Seismological Society of America*, 110(4), 1904-1923

[2] Böse, M., Heaton, T., & Hauksson, E. (2012). Rapid estimation of earthquake source and ground-motion parameters for earthquake early warning using data from a single three-component broadband or strong-motion sensor. *Bulletin of the Seismological Society of America*, 102(2), 738-750.

[3] Picozzi, M., Zollo, A., Brondi, P., Colombelli, S., Elia, L., & Martino, C. (2015). Exploring the feasibility of a nationwide earthquake early warning system in Italy. *Journal of Geophysical Research: Solid Earth*, 120(4), 2446-2465.

3. The authors now consider the saturation of the Early warning indicators with magnitude and they expand the time window, to account for correct magnitude estimation of large events. However, it seems to me that they only assume a variable delay as a function of magnitude, which is not correct for regional Early warning systems. Within this strategy, S wave arrives at close stations well before having 12s or 20s of P wave time window, and those stations need to be not considered for magnitude estimations. Thus, following this simplistic approach, we should only use stations located enough far to catch 12s or 20s of P wave, avoiding the pollution of S waves.

This comment implicitly centres on the performance/accuracy of the underlying EEW algorithm(s), which is not within the scope of this study. We have purposely not mentioned or included any specific algorithm, given that:

- *It is likely that different algorithms would suit different regions;*
- *The work is foremost a feasibility study;*
- *Our paper nevertheless follows the state-of-the-art approach to addressing alert accuracy across a large scale [1];*
- *This type of examination was carried out for select testbed sites across Europe in previous studies by the same authors [2,3].*

We have added a note on this topic to the last paragraph of the “Discussion” section, such that the final section of this paragraph now reads: “Precisely characterising warning accuracy would involve more detailed analysis with the specific algorithms of operational EEW platforms, including the quantification and propagation of uncertainties at each step of the calculations. This type of examination was carried out for select testbed sites across Europe in previous studies by the same authors (42,43). It is outside the scope of this paper, given the continent-wide extent of the study (i.e., it is likely that different EEW algorithms would suit different regions) and the fact that this work is foremost an investigation of feasibility. Finally, we did not consider the economic value of EEW, i.e., the costs required to build and maintain EEW systems compared to the monetary savings they provide through avoided damage (44). Despite these constraints, this study nevertheless represents a first attempt to comprehensively quantify potential EEW effectiveness on a continental scale and to identify priority regions for more detailed EEW feasibility analyses/investment in EEW implementation.”

In any case, inaccurate magnitude estimations due to S-wave contamination would not necessarily lead to a malfunction of the EEW system. For example, if the estimated magnitude of an incoming 7.5 magnitude

earthquake happened to be 8.0, the system would still issue an alarm as required in the likely event that the alarm magnitude threshold was less than or equal to 7.5.

References:

[1] Minson, S. E., Baltay, A. S., Cochran, E. S., Hanks, T. C., Page, M. T., McBride, S. K., ... & Meier, M. A. (2019). The limits of earthquake early warning accuracy and best alerting strategy. *Scientific reports*, 9(1), 1-13.

[2] Cremen, G., Zuccolo, E., & Galasso, C. (2021). Accuracy and Uncertainty Analysis of Selected Methodological Approaches to Earthquake Early Warning in Europe. *Seismological Research Letters*, (in press).

[3] Zuccolo E., Cremen G., Galasso C. (2021). Comparing the performance of regional earthquake early warning algorithms in Europe, *Frontiers in Earth Science/Geohazards and Georisks*, (in press).

4. Finally, I apologize to have provided the impression that I do not appreciate a 1D model. I do appreciate 1D models even for travel-time computation. I just point out that a uniform 1D model – CRUST 1.0 - based on long period arrival times - is not considered as a reliable model for local measurements of travel-times. Regional Early Warning techniques use 1D models, but calibrated for the target area. Thus, a collection of diverse 1D models, for the different tectonic regions, will provide a significantly more accurate velocity model, that would be an effective added value for this study.

We appreciate this comment. However, we again emphasise that this is a continent-wide study, where developing a large collection of local 1D models is not within the scope of the study. As mentioned in our response to Comment #1, we believe that, given the large scale of the study area, it is important to use consistent and open high-level data sources that ensure the results are replicable.

Reviewer #2 (Dr. David Wald):

I give the authors an enormous amount of credit for their successful and comprehensive revision (and suitable rebuttals in a few other cases). I reviewed the reconciliation to the other two reviewers' comments and those too were addressed comprehensively by the authors.

In this case, the peer-review process worked as hoped. The revision is reframed (right down to the title) to the extent that it is a much more balanced and valuable paper than the original. It is ready for publication and will make a valuable contribution to this topic, likely leading the EEW discussion in Europe.

We thank the Reviewer for their positive assessment of our revisions and rebuttals. We would like to reiterate our appreciation for the detailed comments of the Reviewer in the first round of review that ultimately led to a significantly improved manuscript.

There are two minor quibbles that I mention below, that do not require attention at this time, but the authors might mull over:

1. #5 “the average annual European GDP affected by earthquakes exceeds \$20 billion”. I still think that juxtaposing total earthquake losses with EEW implicitly makes the case that those losses could be reduced by successful EEW. I would provide a caveat there.

Thanks for making this point. To address it, we have modified the following line “this is underlined by the fact that the average annual European GDP affected by earthquakes exceeds \$20 billion” to: “EEW could

potentially contribute towards reducing the more than \$20 billion of European GDP that is affected annually by earthquakes (on average)."

2. Item #13, concerning "removal vehicles from garages". Since the study that recommended that gave interviewees 50-sec warning, and asked what they could do with 50 sec, that response came up. It seems that this is no longer a practical consideration given what we've learned in this study so you might consider eliminating that as a plausible mitigation strategy. Going into a garage for this express purpose seems like a bad idea.

We thank the Reviewer for this comment and, as a result, have removed reference to the highlighted action in the paper.

Reviewer #3 (Dr. Yannick Behr):

The manuscript with the title "Could earthquake early warning be effective across Europe" has much improved by the significant amount of work the authors put in to address the reviewers comments. Reported lead times seem more realistic and, by adding additional magnitude dependent delays, are more in line with recent EEW research. Also, many assumptions underlying the lead time calculations have been made more explicit. Overall the text reads well and has become more concise. I particularly welcome the fact that the authors are planning to publish the source code of their analysis.

We really appreciate the Reviewer's positive assessment of our revised manuscript.

1. The updated version has, however, increased my doubts about the usefulness of a relative feasibility index as presented by the authors. Although median lead times have decreased significantly in many regions, the feasibility index for regions with positive median lead times appears unchanged or even improved. I just do not understand how such an index could be useful to decision makers who have to decide whether or not to invest in EEW. To me it masks the fact that median lead times are, in the majority of the cases, between 0-5 s. In my opinion, deciding on an, albeit subjective, lead time that would be 'useful' and then measuring the feasibility based on that would provide a lot more value. If the authors decide to keep using the relative index I think they should make it crystal clear how to interpret the index. The easiest way to do that would be by adding a few examples for various indices (e.g. 1, 0.8, 0.2). The authors added an additional, useful, explanation to the Methods section, but I think a few examples with actual values in the main text are needed too.

We thank the Reviewer for this comment. Each component of the relative feasibility index (respectively related to lead time, population, and seismic hazard) provides a relative rank (score) for all considered sites. This means that the site with the highest lead time will receive the same lead-time score (1), regardless of whether the associated maximum lead-time value is 100 or 1 seconds. The relative scoring system of the feasibility index explains why the index for regions with positive median lead times "appears unchanged or even improved".

It is important to note that the relative feasibility index itself does not "mask the fact that median lead times are, in the majority of the cases, between 0-5 s"; we can account for the "non-feasibility" of sites with less than 5 seconds of lead time in the relative feasibility index calculation if required, by simply removing these sites from consideration. However, we have decided to calculate the relative feasibility index for all sites with median lead times greater than 0 seconds, because positive lead times less than 5 seconds are still useful; for example, per Table 2 of the manuscript, these times can facilitate automatic actions like the switching of traffic lights.

We thank the Reviewer for the suggestion to better explain the meaning of the index in the main text. We have implemented this suggestion, by adding the following text to the "EEW Feasibility Calculation" subsection of Results: "For context, a site associated with the 10th percentile value of positive L, the 20th percentile value of I, and the 40th percentile value of P would yield the following relative feasibility indices for the different

examined weighting strategies: 0.23 ($w_P = w_I = w_L = 0.333$), 0.18 ($w_L = 0.6$, and $w_P = w_I = 0.2$), 0.22 ($w_I = 0.6$, and $w_L = w_P = 0.2$), and 0.3 ($w_P = 0.6$, and $w_L = w_I = 0.2$)". We have also added a note that the relative feasibility index ranges in value from 0 to 1, in the introductory sentence of this subsection.

2. My other main point may seem like a technicality but I think it is quite important: with the updated lead times there is even more reason to focus the maps on the regions that are actually discussed in the manuscript. For example, Portugal shows a small area with positive median lead times but this is never mentioned in the text. In the updated analysis, Iceland doesn't have positive lead times anymore. I think it suffices to mention that the analysis was done for the whole Euro-Mediterranean region but that only countries will be discussed for which EEW seems feasible. There is a lot of wasted space in these maps which makes it also quite difficult to see details in the parts of the maps that matter.

This is a good point. To address it, we have now focused the relative feasibility index maps on the Southern part of Europe (please see an example map in Figure 3). However, we have kept the entire Euro-Mediterranean extents of the lead-time maps because we believe that geographical visualisations of regions with negative lead times are informative for the reader and are important to the overall findings of the study.

We acknowledge that Portugal is not explicitly mentioned in the text as a region with positive median lead time. Indeed, many other countries with positive median lead time (such as Slovenia, Croatia, and Bugaria) are also not referenced in the text. This is because we believe it is more effective to instead provide a visual summary of all regions with positive median lead times in a figure, which is the purpose of the paper's Figure 4 (b).

Figure 3: Example of the modified Relative Feasibility Index mapping included in the revised version of the manuscript.

Reviewers' comments:

Reviewer #1 (Remarks to the Author):

we are now at the third round of revisions of the manuscript and important, critical issues (in my opinion) raised from the first round of revisions are still there.

The station distribution is still not realistic, because the database of stations selected by the authors is not appropriate for the purpose of this study. This database maps mostly broad-band stations, and national-wide (not small scale) strong motion networks are missing. Missing information is expected to completely modify the station density and distribution, affecting the results and the conclusions. I'm sensible to the reliability of the databases, of course, but in my opinion they used the wrong one. They could have checked and worked with the European Strong Motion database, which is indeed appropriate for this study, <https://esm.mi.ingv.it/> and contains many of the strong-motion national networks.

Further, EEW are based on reliable strong-motion recording. You can ask any seismologist, if he will invest rather in a broad-band or in accelerometric network for strong motion recording. Thus, the applicability of broad-band stations for analysis and recording of strong motion from (very) large earthquakes is a fundamental issue of this study, significantly limiting the explored magnitude/distance range. Also, this issue is well known in the ShakeAlert system, cited by the authors (Kohler et al., 2018), where they clearly state (Pag.3) "EEW seismometer sensor types include broadband, short period and strong motion. Broadband sensors typically record velocity time series and are the most sensitive to low-amplitude motions (i.e., producing high signal-to-noise time series) for the wide range of seismic frequencies, but they can clip for large amplitudes", later on they declare that this is a limitation!

Finally, about methodology, I strongly disagree with the authors. As a general feature, S waves pollute the P wave if the P wave time window to be selected is long (10-12s), to maintain a general approach, you should discard traces that include the S waves. Only if considering specific algorithms, one could try to filter out S waves. Thus, adopting a general approach for EEW and considering 10s of P waves, data recorded at distances shorter than 80-90 km from the source should not be considered.

I thus am still skeptical about the validity of the conclusions.

Reviewer #3 (Remarks to the Author):

I am happy with the changes the authors made to Figure 8 and also agree with their choice to display the larger map area in the other figures.

I still disagree, however, with the usefulness of a relative feasibility index as I think it is misleading or at least overoptimistic. The authors' reply in the rebuttal letter was disappointing as they merely explained again the index rather than providing a stronger argument for using it. The index will be particularly problematic if the cumulative distribution functions are steep. In such a case, a high and low index value may only be separated by a very small difference in the underlying metric (lead time, average seismic intensity, affected ambient population).

Another problem is that it is insensitive to constant offsets. If, for example, the lead time was reduced by a few seconds at every site, the index would only change for sites where lead times became negative. In fact, I suspect that is the reason why the index did not change significantly when the triggering PGA was set to 0.02, 0.1 or 0.2 g.

If the authors decided to keep the relative index, I think they have to make it very clear what it means. I already wrote this in my previous review and I don't think the sentence added by the authors is sufficient. The minimum would be to show two cases with different indices and the underlying metrics. Taking the newly added sentence as a starting point it could be something like:

"For context, a site associated with ?? s of positive lead time (10th percentile), ?? of average intensity (10th percentile), and ?? of ambient population (10th percentile) would yield the following relative feasibility indices for the different examined weighting strategies: [...]. In contrast a site with ?? s of positive lead time (90th percentile), ?? of average intensity (90th percentile), and ?? of ambient population (90th percentile) would yield the following relative feasibility indices for the different examined weighting strategies: [...]."

Even better would be to also show the underlying ECDFs. I would also recommend to provide a

rationale for the design of the index.

I think the rest of the manuscript is ready for publication.

Yannik Behr

Response to comments for "Investigating the potential effectiveness of earthquake early warning across Europe", by Gemma Cremen, Carmine Galasso, and Elisa Zuccolo

NCOMMS-20-42346-R3

We thank the reviewers for their thoughtful comments, which have further improved the quality of the revised manuscript. The reviewer's comments have been numbered and listed below, followed by our responses in italics.

Reviewer #1:

We are now at the third round of revisions of the manuscript and important, critical issues (in my opinion) raised from the first round of revisions are still there.

We thank the reviewer for their additional comments, which are addressed in detail below.

1. The station distribution is still not realistic, because the database of stations selected by the authors is not appropriate for the purpose of this study. This database maps mostly broad-band stations, and national-wide (not small scale) strong motion networks are missing. Missing information is expected to completely modify the station density and distribution, affecting the results and the conclusions. I'm sensible to the reliability of the databases, of course, but in my opinion they used the wrong one. They could have checked and worked with the European Strong Motion database, which is indeed appropriate for this study, <https://esm.mi.ingv.it/> and contains many of the strong-motion national networks.

*We are surprised that the reviewer has not mentioned or suggested using the European Strong Motion (ESM) database in the **two** previous rounds of review.*

We have now extracted this database, discarding all stations that were removed before January 2021 and all stations that were installed after this date (to be consistent with the approach to station selection used for the IRIS database). The spatial distribution of ESM stations is presented in Figure 1, as well as the spatial distribution of IRIS stations used in our study.

We respectfully disagree with the reviewer's opinion that the ESM database is more appropriate for this study, for the following reasons:

- (1) The ESM database does not contain any seismic stations located in Iceland (see Figure 1), which is one of the countries selected for inclusion in our study based on its high seismic hazard (see Figure 2 of the manuscript). On the other hand, the IRIS database used in our study contains seismic stations across all countries selected for inclusion in our analyses. When faced with a choice of two databases, one that covers all countries of interest and one that does not, it seems appropriate to us to choose the former.*
- (2) The ESM database is not a formal repository of seismic networks, but a collection of processed waveforms for events that occur in the Euro-Mediterranean and the Middle East. On the other hand, the IRIS database contains complete information on seismic stations associated with the International Federation of Digital Seismograph Networks (FDSN). According to the FDSN website homepage (<https://www.fdsn.org/about/>), "members agree to coordinate station siting and provide free and*

open access to their data.. FDSN goals related to station siting and instrumentation are to provide stations with good geographic distribution..” It seems appropriate to us to use a formal database of seismic stations that has been developed with explicit consideration of station siting, over a library of processed waveforms in which station information is secondary data.

In any case, the databases possess a number of similarities:

- (1) Both databases contain an approximately equivalent number of stations (there are 2,377 stations in the study database versus 2,299 in the the ESM database)*
- (2) The median interstation distance across target sites considered in the feasibility analyses of the study is 25 km for the study database and 23 km for the ESM database, while the corresponding average values are respectively 27 km and 24 km. Interstation distances differ between the databases by less than 10 km for 75% of these target sites.*

These similarities do not suggest that there are clear advantages of choosing the ESM database over the database currently used in the study.

Considering all the above points (and the fact that the IRIS database is open source, ensuring the study is fully replicable) we have decided to keep the station database as is.

Figure 1: Spatial distribution of seismic stations found in the ESM database. Also shown are the IRIS database seismic stations used in our study.

2. Further, EEW are based on reliable strong-motion recording. You can ask any seismologist, if he will invest rather in a broad-band or in accelerometric network for strong motion recording. Thus, the applicability of broad-band stations for analysis and recording of strong motion from (very) large earthquakes is a fundamental issue of this study, significantly limiting the explored magnitude/distance range. Also, this issue is well known in the ShakeAlert system, cited by the authors (Kohler et al., 2018), where they clearly state (Pag.3) “EEW seismometer sensor types include broadband, short period and strong motion. Broadband sensors typically record velocity time series and are the most sensitive to low-amplitude motions (i.e., producing high signal-to-noise time series) for the wide strange of seismic frequencies, but they can clip for large amplitudes”, later on they declare that this is a limitation!

The reviewer’s comment points to the use of broadband stations in existing EEW systems and as such, supports our decision for including broadband stations in our analyses. Note that we cannot find a discussion on the limitations of using broadband stations in the paper mentioned by the reviewer.

Also, the third author of the study is a seismologist (and **she** advised on this issue). In particular, we are all well aware of the limitations of broadband sensors and of the fact that broadband recordings saturate in the proximity of the source. Nevertheless - as mentioned in previous rounds of review – this is a feasibility study and we clearly state our working assumption is that all considered stations will be upgraded for actual use in earthquake early warning if necessary (see Discussion section): “it is assumed that the considered seismic stations are (or could be) capable of being used for early warning purposes (i.e., they have or could have adequate data acquisition/transmission systems, real-time communication capability, robust dissemination methods, power supply systems, etc.)”.

3. Finally, about methodology, I strongly disagree with the authors. As a general feature, S waves pollute the P wave if the P wave time window to be selected is long (10-12s), to maintain a general approach, you should discard traces that include the S waves. Only if considering specific algorithms, one could try to filter out S waves. Thus, adopting a general approach for EEW and considering 10s of P waves, data recorded at distances shorter than 80-90 km from the source should not be considered.

We have addressed this comment, by investigating the effect on the calculations of removing from consideration all sources for which the magnitude computation time is assumed to be greater than 10 seconds and the closest source to station distance is less than 80km. Figure 2 displays the difference in median, maximum, and minimum leadtimes obtained for sites at which this difference is at least one second and the original and/or modified leadtimes are positive. Table 1 displays the percentage of all considered target sites at which the difference is at least one and five seconds, and the original and/or modified leadtimes are positive. It can be seen from both Figure 2 and Table 1 that the number of affected target sites is small. In addition, the conclusions of the lead-time calculations are not significantly affected:

- 4% (rather than 3%) of minimum lead times are positive
- 21% (rather than 18%) of median lead times are positive
- 75% (rather than 79%) of maximum lead times are positive
- The maximum lead time achieved across all target sites is 15.5s (rather than 17.2s)

Table 1: Percentages of considered target sites at which lead-time changes exceed one and five seconds (and the original and/or modified leadtimes are positive), across the different lead-time maps.

Lead Time Map	Percentage of sites with lead-time difference greater than one second	Percentage of sites with lead-time difference greater than five seconds
Minimum	1.1%	1.0%
Median	5.1%	2.1%
Maximum	12.4%	4.1%

(a)

(b)

(c)

Figure 2: Differences in median, maximum, and minimum leadtimes obtained if we remove from the analyses sources for which the magnitude computation is assumed to be at least 10 seconds and the minimum source to station distance is less than 80 km. We only highlight sites for which the leadtime difference is at least one second and the original and/or modified leadtimes are positive. Blue colours indicate larger lead times in the original analyses, whereas red colours denote larger lead times in the modified calculations.

We have additionally investigated the effect on the feasibility calculations; see Figure 3, where the left-hand column provides the feasibility results for the original calculations and the right-hand column provides these results for the modified calculations. It can be inferred from this figure that the conclusions of the feasibility calculations do not change. That is, the most feasible regions for EEW are still predominantly found in Italy, Turkey, and Greece.

Figure 3: Indices for (a-b) the case in which lead time, intensity, and population are equally weighted by a stakeholder, as well as differences for cases in which (c-d) lead time, (e-f) seismic intensity, and (g-h) population are respectively weighted three times more than both other variables. Also shown are (i-j) equally weighted indices modified in line with the relative number of correctly issued alerts (for a 0.05 g alert threshold). The left-hand column contains the results of the original calculation and the right-hand column contains the results of the modified calculations.

We accept the reviewer’s viewpoint that by accounting for long P-wave windows in our calculations, we are making an implicit assumption that the underlying EEW algorithm is capable of filtering out polluting S-waves. However, given the above investigation, we can conclude that this assumption does not have a significant effect on the results. We have clarified these points in the “Lead-time Modelling” subsection of the “Methods” section, as follows: “Note that the relatively large δ_m values for magnitudes greater than or equal to 7 require an implicit assumption that the underlying EEW algorithm is capable of filtering out polluting S-waves from long P-wave windows. The validity of this assumption does not significantly affect the outcomes of the study, however; removing from the analyses sources for which $\delta_m \geq 12$ and the nearest station is less than 80 km does not change the conclusions of the work.”

Reviewer #3:

I am happy with the changes the authors made to Figure 8 and also agree with their choice to display the larger map area in the other figures.

I still disagree, however, with the usefulness of a relative feasibility index as I think it is misleading or at least overoptimistic. The authors’ reply in the rebuttal letter was disappointing as they merely explained again the index rather than providing a stronger argument for using it. The index will be particularly problematic if the cumulative distribution functions are steep. In such a case, a high and low index value may only be separated by a very small difference in the underlying metric (lead time, average seismic intensity, affected ambient population).

We apologise for disappointing the reviewer with our previous response. We now plot the empirical cumulative distribution functions of median lead time (L), average seismic intensity (I), and ambient population (P) in Figure 4. It can be seen from this figure that none of the cumulative distributions are particularly steep, thus the problem perceived by the reviewer does not feature in this study. In any case, the purpose of the index (similar to many other decision-making algorithms/tools) is to select the most feasible regions, regardless of the extent of the difference between their feasibility and less feasible regions.

In any case, it is not clear how an absolute version of the feasibility index would work without some type of normalisation of the different variable values, to facilitate their ultimate combination as one metric. This would require either:

- (1) Normalising the values relative to each other (e.g., min-max normalisation), similar to the underlying process of the Relative Feasibility Index proposed in this study; or

(2) Normalising the values relative to some “ideal” value (e.g., 10 seconds of lead time for L), which would require a subjective (therefore arbitrary and potentially problematic) choice by the modeller on what constitutes this ideal value for each variable.

Figure 4: Empirical cumulative distribution functions of (a) median lead time (L), (b) average seismic intensity (I), and (c) ambient population (P)

Another problem is that it is insensitive to constant offsets. If, for example, the lead time was reduced by a few seconds at every site, the index would only change for sites where lead times became negative. In fact, I suspect that is the reason why the index did not change significantly when the triggering PGA was set to 0.02, 0.1 or 0.2 g.

The insensitivity of the feasibility index to constant offsets is actually an advantage - rather than a shortcoming- of the proposed index. This means that the index can be robust to potentially arbitrary assumptions about consistent EEW transmission/telemetry delays.

It is important to mention that alert accuracy is computed completely independently of the lead-time calculations. As implied by equation 4 of the methods section, alert accuracy is simply added on to the pre-existing relative feasibility index calculation (and the weights are modified accordingly). Thus, lead-time differences do not explain similarities in alert-accuracy ranking for different PGA thresholds. In addition, it is important to note that while the countries with the highest feasibility remain the same when the PGA thresholds are changed, there are spatial variations in the most feasible sites within these countries.

If the authors decided to keep the relative index, I think they have to make it very clear what it means. I already wrote this in my previous review and I don't think the sentence added by the authors is sufficient. The minimum would be to show two cases with different indices and the

underlying metrics. Taking the newly added sentence as a starting point it could be something like: “For context, a site associated with ?? s of positive lead time (10th percentile), ?? of average intensity (10th percentile), and ?? of ambient population (10th percentile) would yield the following relative feasibility indices for the different examined weighting strategies: [...]. In contrast a site with ?? s of positive lead time (90th percentile), ?? of average intensity (90th percentile), and ?? of ambient population (90th percentile) would yield the following relative feasibility indices for the different examined weighting strategies: [...].”

We thank the reviewer for this suggestion, which we have implemented by adding a second case. The two cases now read in the text as follows:

“Note that a site associated with the 10th percentile value of positive L, the 20th percentile value of I, and the 40th percentile value of P would yield the following relative feasibility indices for the different examined weighting strategies: 0.23 ($w_p = w_I = w_L = 0.333$), 0.18 ($w_L = 0.6$, and $w_p = w_I = 0.2$), 0.22 ($w_I = 0.6$, and $w_L = w_p = 0.2$), and 0.3 ($w_p = 0.6$, and $w_L = w_I = 0.2$)”. In contrast, for the same respective weighting strategies, a site associated with the 90th percentile value of positive L, the 80th percentile value of I, and the 60th percentile value of P would produce relative feasibility indices of 0.77, 0.82, 0.78, and 0.7.”

Note that we did not use equivalent percentiles for each variable in our example cases (as implied by the reviewer’s suggestion), as this would lead to identical feasibility indices across all weighting strategies and therefore is not ideal for illustrating its sensitivities.

Even better would be to also show the underlying ECDFs. I would also recommend to provide a rationale for the design of the index.

Thanks for these suggestions. We have added Figure 4 to the paper (as Figure 8), and explained it at the end of the first paragraph in the “EEW Feasibility Calculation” section of Results, as follows: “For context, Figure 8 provides the empirical cumulative distribution functions of L, P, and I that are used to derive the index.”

We have also provided a rationale for the design of the index at the end of the first paragraph in the “EEW Feasibility Calculation” section of Results, as follows:

“Note that the purpose of the index is to identify the most feasible regions for EEW, regardless of the extent to which their feasibility differs to that of less feasible regions (and therefore the steepness of the underlying empirical cumulative distribution functions). This approach is consistent with many multi-criteria decision-making tools, including TOPSIS (the Technique for Order of Preference by Similarity to Ideal Solution) [1] and multi-attribute utility theory [2].

References:

[1] Edwards, W., & Newman, J. R. (1986). Multiattribute evaluation. Cambridge University Press.

[2] Yoon, K. P., & Hwang, C. L. (1995). Multiple attribute decision making: an introduction. Sage publications.

I think the rest of the manuscript is ready for publication.

We thank the reviewer for their positive assessment of our work.

Yannik Behr

REVIEWERS' COMMENTS

Reviewer #3 (Remarks to the Author):

I would like to thank the authors for their continued efforts to address the reviewers' comments. I am still not convinced by the usefulness of the feasibility index. It is my belief that "Normalising the values relative to some "ideal" value (e.g., 10 seconds of lead time for L)," , albeit dependent on subjective choices, would be a lot more useful. Ultimately, a feasibility index is meant to help decision makers to decide if, where, and how to invest in EEW. An index that does not change significantly when the underlying parameters are all reduced by the same amount does not seem very helpful in this context. However, since this is coming down to a matter of opinion and the authors have provided enough detail for the reader to make an informed decision about the index' usefulness, I am happy for it to be published. After all, scientific publications are a way of scientific discourse rather than a statement of consensus.

Kind regards
Yannik Behr

Reviewer #4 (Remarks to the Author):

The manuscript makes an attempt to evaluate the possible utility (or at least the relative utility) of EEW implementation across Europe. An assessment of this kind on a scale like this necessitates many assumptions. The question for this manuscript is, do the assumptions necessary render the conclusions meaningless? I believe the conclusions are valid and reasonable. I also think the authors have done a good job of responding to the reviewer comments and in modifying the manuscript accordingly. Therefore I recommend the manuscript now be published.

Like the reviewers, I am not a fan of the relative feasibility index. My criticism is not of the presented index specifically, but rather of the whole concept that we have to combine multiple parameters (with different units) into a single value using arbitrary weighting. In reality there is a complex interplay between these variables when it comes to the question of delivering usable alerts. Still, the overall outcome of the analysis is to highlight where EEW would be most useful and where it would be less useful. As long as these assessments are only seen as being relative value I think it is useful which is why I am recommending publication.

Perhaps the key conclusion from the paper is that the three countries which the greatest potential to benefit from EEW are Greece, Turkey and Italy. This intuitively makes sense and is clearly born-out by the analysis presented. I note that two of these countries now have nation-wide EEW (this should be noted in the manuscript). Greece and Turkey are part of the Android Earthquake Alert System which is using Android phones to detect earthquakes and deliver alerts (also to Android phones). See <https://blog.google/products/android/new-features-summer-2021/>

Reviewer 1

Comment 1: which station database to use.

The authors choice of station database is reasonable.

Comment 2: use of strong motion vs broadband data

Broadband stations absolutely can be used for EEW; we use all of them in California.

Comment 3: possible pollution of P-waves by S-waves for long source durations

The authors have shown that removing these cases does not significantly change their conclusions. Also, EEW system typically do include some capability to recognize and remove S-wave contamination from P-wave signals limiting this problem.

Reviewer 3

These comments are all about the limitations of the metric the authors have designed and used.

As described above, no metric is ideal, and the authors choice is not unreasonable. They have also carefully detailed how they have calculated it, so I think this is reasonable.

Richard Allen - UC Berkeley

Response to comments for "Investigating the potential effectiveness of earthquake early warning across Europe", by Gemma Cremen, Carmine Galasso, and Elisa Zuccolo

NCOMMS-20-42346-R4

We thank the reviewers for their thoughtful comments, which have further improved the quality of the revised manuscript. The reviewer's comments have been numbered and listed below, followed by our responses in italics. Extracts from the text are coloured blue.

Reviewer #3 (Dr. Yannick Behr):

I would like to thank the authors for their continued efforts to address the reviewers' comments. I am still not convinced by the usefulness of the feasibility index. It is my belief that "Normalising the values relative to some "ideal" value (e.g., 10 seconds of lead time for L)," , albeit dependent on subjective choices, would be a lot more useful. Ultimately, a feasibility index is meant to help decision makers to decide if, where, and how to invest in EEW. An index that does not change significantly when the underlying parameters are all reduced by the same amount does not seem very helpful in this context. However, since this is coming down to a matter of opinion and the authors have provided enough detail for the reader to make an informed decision about the index' usefulness, I am happy for it to be published. After all, scientific publications are a way of scientific discourse rather than a statement of consensus.

We thank Dr. Behr for his patience throughout the review process, and for all of his extremely insightful comments that have helped to significantly improve the quality of the revised manuscript.

Reviewer #4 (Prof. Richard Allen):

The manuscript makes an attempt to evaluate the possible utility (or at least the relative utility) of EEW implementation across Europe. An assessment of this kind on a scale like this necessitates many assumptions. The question for this manuscript is, do the assumptions necessary render the conclusions meaningless? I believe the conclusions are valid and reasonable. I also think the authors have done a good job of responding to the reviewer comments and in modifying the manuscript accordingly. Therefore I recommend the manuscript now be published.

We thank Prof. Allen for his review and his positive assessment of our work. We address each comment below individually.

Like the reviewers, I am not a fan of the relative feasibility index. My criticism is not of the presented index specifically, but rather of the whole concept that we have to combine multiple parameters (with different units) into a single value using arbitrary weighting. In reality there is a complex interplay between these variables when it comes to the question of delivering usable alerts. Still, the overall outcome of the analysis is to highlight where EEW would be most useful

and where it would be less useful. As long as these assessments are only seen as being relative value I think it is useful which is why I am recommending publication.

We thank the reviewer for this comment, and for confirming the usefulness of the relative feasibility index metric. We have further emphasised that the feasibility index is relative in the newest version of the manuscript, by adding the description “relative” to every mention of/reference to the feasibility index in the “Discussion” section.

Perhaps the key conclusion from the paper is that the three countries which the greatest potential to benefit from EEW are Greece, Turkey and Italy. This intuitively makes sense and is clearly born-out by the analysis presented. I note that two of these countries now have nation-wide EEW (this should be noted in the manuscript). Greece and Turkey are part of the Android Earthquake Alert System which is using Android phones to detect earthquakes and deliver alerts (also to Android phones). See

<https://blog.google/products/android/new-features-summer-2021/>

We thank the reviewer for pointing out that the Android Earthquake Alert System is now operational in both Turkey and Greece. This has been noted in the revised version of the manuscript in both the “Introduction” and the “Discussion” sections, as follows:

- 1. The first part of the third sentence in the penultimate paragraph of the “Introduction” section now reads: “The only European countries with current government-supported operational EEW systems are Romania [1] and Turkey [2] (the Android Earthquake Alert System, which uses Android phones to issue and received early warnings [3-6], has also recently been launched in Greece as well as Turkey [7]), despite a strong need to develop effective measures for mitigating seismic risk across many parts of the continent...”*
- 2. The first part of the penultimate paragraph in the “Discussion” section now reads: “In particular, the computed relative feasibility indices suggest that an expansion of permanent EEW efforts in Turkey beyond Istanbul (by upgrading the hardware and software of existing strong-motion/broadband stations and networks for real-time data processing and telemetry capabilities) could be appropriate, supporting the recently launched Android Earthquake Alert System in the country. The promising results of the relative feasibility mapping for Italy and Greece are particularly notable, since neither has a current permanently operational EEW system (although Greece is now also benefiting from the Android Earthquake Alert System).”*

References :

*[1] Mărmureanu, A., Ionescu, C., & Cioflan, C. O. (2011). Advanced real-time acquisition of the Vrancea earthquake early warning system. *Soil Dynamics and Earthquake Engineering*, 31(2), 163-169.*

*[2] Alcik, H., Ozel, O., Apaydin, N., & Erdik, M. (2009). A study on warning algorithms for Istanbul earthquake early warning system. *Geophysical Research Letters*, 36(5).*

*[3] Kong, Q., Martin-Short, R., & Allen, R. M. (2020). Toward Global Earthquake Early Warning with the MyShake Smartphone Seismic Network, Part 1: Simulation Platform and Detection Algorithm. *Seismological Research Letters*, 91(4), 2206-2217.*

[4] Kong, Q., Martin-Short, R., & Allen, R. M. (2020). Toward Global Earthquake Early Warning with the MyShake Smartphone Seismic Network, Part 2: Understanding MyShake Performance around the World. *Seismological Research Letters*, 91(4), 2218-2233.

[5] Allen, R. M., Kong, Q., & Martin-Short, R. (2020). The MyShake platform: a global vision for earthquake early warning. *Pure and Applied Geophysics*, 177(4), 1699-1712.

[6] Cardno, C. A. (2020). Android Phones Now Offer Earthquake Detection, Alerts. *Civil Engineering Magazine Archive*, 90(10), 34-36.

[7] Lee, F. (2021). 6 New Features on Android This Summer. *The Key Word*. <https://blog.google/products/android/new-features-summer-2021/>

Reviewer 1

Comment 1: which station database to use.

The authors choice of station database is reasonable.

We thank the reviewer for making this point, which supports our selection of seismic stations.

Comment 2: use of strong motion vs broadband data

Broadband stations absolutely can be used for EEW; we use all of them in California.

We thank the reviewer for making this point, which supports our previous responses to Reviewer #1 on the topic of broadband stations.

Comment 3: possible pollution of P-waves by S-waves for long source durations

The authors have shown that removing these cases does not significantly change their conclusions. Also, EEW system typically do include some capability to recognize and remove S-wave contamination from P-wave signals limiting this problem.

We thank the reviewer for this comment, which supports our use of an implicit assumption that any implemented EEW algorithms will be capable of filtering out polluting S-waves from long P-wave windows. In any case, as the reviewer has pointed out, our conclusions are not sensitive to this assumption.

Reviewer 3

These comments are all about the limitations of the metric the authors have designed and used.

As described above, no metric is ideal, and the authors choice is not unreasonable. They have also carefully detailed how they have calculated it, so I think this is reasonable.

We thank the reviewer for this comment, which supports our use of the relative feasibility index. We have further emphasised that the feasibility index is relative in the newest version of the manuscript, by adding the description "relative" to every mention of/reference to the feasibility index in the "Discussion" section.